# Cathelicidins prime platelets to mediate arterial thrombosis and tissue inflammation

Joachim Pircher[1,2], Thomas Czermak[1], Andreas Ehrlich[1], Clemens Eberle[1], Erik Gaitzsch[3], Andreas Margraf[3], Jochen Grommes [4,5], Prakash Saha[6], Anna Titova[1], Hellen Ishikawa-Ankerhold[1], Konstantin Stark[1,2], Tobias Petzold[1,2], Thomas Stocker[1], Ludwig T Weckbach[1,3], Julia Novotny[1], Markus Sperandio[2,3], Bernhard Nieswandt[7,8], Alberto Smith[6], Hanna Mannell[3], Barbara Walzog [3], David Horst[9], Oliver Soehnlein[2,4,10], Steffen Massberg[1,2] & Christian Schulz [1,2]

Leukocyte-released antimicrobial peptides contribute to pathogen elimination and activation of the immune system. Their role in thrombosis is incompletely understood. Here we show that the cathelicidin LL-37 is abundant in thrombi from patients with acute myocardial infarction. Its mouse homologue, CRAMP, is present in mouse arterial thrombi following vascular injury, and derives mainly from circulating neutrophils. Absence of hematopoietic CRAMP in bone marrow chimeric mice reduces platelet recruitment and thrombus formation. Both LL-37 and CRAMP induce platelet activation in vitro by involving glycoprotein VI receptor with downstream signaling through protein tyrosine kinases Src/Syk and phospholipase C. In addition to acute thrombosis, LL-37/CRAMP-dependent platelet activation fosters platelet–neutrophil interactions in other inflammatory conditions by modulating the recruitment and extravasation of neutrophils into tissues. Absence of CRAMP abrogates acid-induced lung injury, a mouse pneumonia model that is dependent on platelet–neutrophil interactions. We suggest that LL-37/CRAMP represents an important mediator of platelet activation and thrombo-inflammation.

[1] Medizinische Klinik und Poliklinik I, Klinikum der Universität München, Ludwig-Maximilians-University, 81377 Munich, Germany. [2] DZHK (German Center for Cardiovascular Research), partner site Munich Heart Alliance, 80802 Munich, Germany. [3] Walter-Brendel-Centre of Experimental Medicine, Department of Cardiovascular Physiology and Pathophysiology, Biomedical Center, Ludwig-Maximilians-University, Planegg-Martinsried, 82152 Munich, Germany. [4] Institute for Cardiovascular Prevention (IPEK), Ludwig-Maximilians-University, 80336 Munich, Germany. [5] Klinik für Gefäßchirurgie, Uniklinik RWTH, 52074 Aachen, Germany. [6] Academic Department of Vascular Surgery, King's College London, St. Thomas' Hospital, London SE1 7EH, UK. [7] Department of Experimental Biomedicine, University Hospital Würzburg, 97080 Würzburg, Germany. [8] Rudolf Virchow Center, University of Würzburg, 97080 Würzburg, Germany. [9] Pathologisches Institut, Ludwig-Maximilians-University, 80337 Munich, Germany. [10] Department of Physiology and Pharmacology, Karolinska Institutet, 17177 Stockholm, Sweden. These authors contributed equally: Joachim Pircher, Thomas Czermak. Correspondence and requests for materials should be addressed to C.S. (email: christian.schulz@med.uni-muenchen.de)

Platelets play a fundamental role in hemostasis. They also contribute to inflammatory conditions and modulate the host immune response[1,2], thereby representing an important link between innate immunity and thrombosis[3,4]. Platelets bind to circulating leukocytes and foster their recruitment to the inflamed or injured vessel wall[5–7]. These interactions, particularly with neutrophils, play an important role in the pathophysiology of common conditions, such as acute lung injury (ALI), ischemic stroke, and organ ischemia-reperfusion injury[8,9]. Previous studies have addressed the question of how platelets affect leukocyte functions and recruitment[10,11], but the reciprocal effects of

leukocytes on platelets and the implications for thrombo-inflammatory processes remain largely elusive.

Innate immune cells actively participate in thrombotic processes. Through initiation of blood coagulation they can induce local thrombosis, which contributes to the containment of pathogens and represents a central mechanism in host defense[4]. Immunothrombosis has long been considered a sole feature of the microvasculature; however, leukocytes are also abundant in arterial thrombi of patients with acute myocardial infarction (AMI)[12,13]. Blood leukocytes, together with platelets, accumulate rapidly at sites of arterial injury[14,15], where activated neutrophils

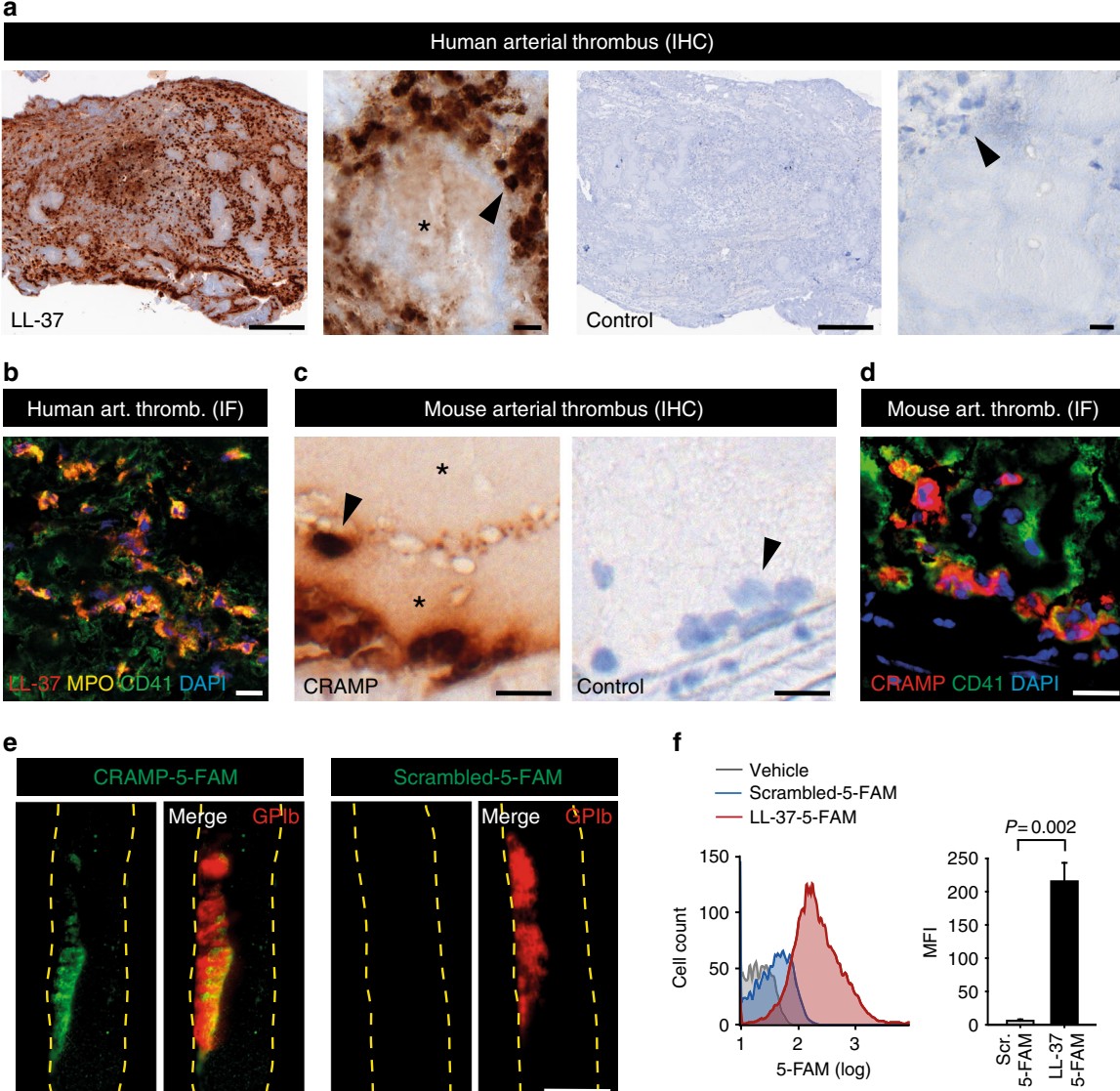

**Fig. 1** Cathelicidins are present in human and mouse arterial thrombi. **a, b** Representative images of coronary artery thrombi isolated from five patients with acute myocardial infarction. **a** Immunohistochemistry for LL-37 indicated enrichment within leukocytes (arrowhead), but also stained leukocyte-free areas (asterisk). Bars, 200 μm (overview) and 10 μm (magnification). **b** Immunofluorescence analysis of LL-37 (red), myeloperoxidase (MPO, yellow), CD41 (platelets, green), and DAPI (nuclei, blue). Bar, 10 μm. **c, d** Representative images of murine carotid artery thrombi generated by ferric chloride injury. **c** Immunohistochemistry for cathelicidin-related antimicrobial peptide (CRAMP, mouse homologue for LL-37) indicated enrichment within leukocytes (arrowhead), but also stained leukocyte-free areas (asterisk). Bars, 10 μm. **d** Immunofluorescence analysis of CRAMP (red), CD41 (platelets, green), and DAPI (nuclei, blue). Bar, 10 μm. **e** Analysis of CRAMP binding in arterial thrombosis in vivo. 5-FAM-labeled CRAMP or scrambled control was injected into wild type mice before induction of ferric chloride injury. Platelets were labeled in vivo using a DyLight649-labeled non-blocking GPIbβ antibody. Left: 5-FAM-labeled CRAMP (green) associated with platelets (GPIb, red in merged image) in the forming thrombus. Right: Image for 5-FAM-labeled control peptide and platelets (GPIb, red in merged image). Bar, 500 μm. See also Supplemental Movies 1, 2. **f** Flow cytometry analysis of LL-37 binding to isolated human, platelets in vitro. 5-FAM-labeled LL-37 (red), scrambled 5-FAM-labeled control peptide (blue), or vehicle (gray). Graph shows mean and SEM. P-value was determined by unpaired t-test

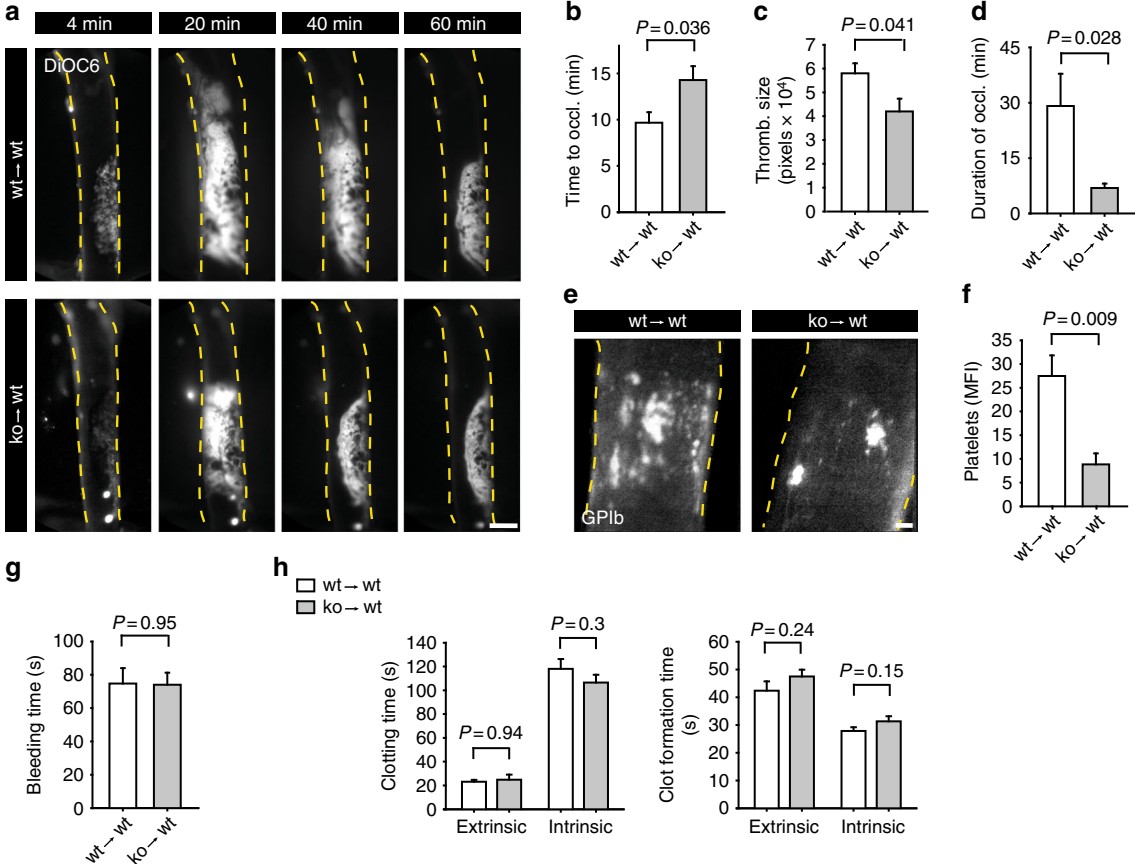

**Fig. 2** Cathelicidins contribute to arterial thrombosis. **a–d** Ferric chloride induced thrombus formation in the mouse carotid artery. **a** Representative intravital microscopy images of wild type (wt → wt) and CRAMP-deficient (ko → wt) BM chimeras. Thrombus size was visualized by in vivo staining with DiOC6. Bar, 400 μm. Analysis of **b** time to complete thrombotic occlusion of the carotid artery, **c** maximal thrombus size, and **d** duration of vessel occlusion ($n = 6$). **e, f** Platelet recruitment and aggregate formation after carotid artery injury induced by temporary mechanical ligation in wild type (wt → wt) and CRAMP-deficient (ko → wt) BM chimeras. Platelets were labeled in vivo using a DyLight488-labeled non-blocking GPIbβ antibody. **e** Representative intravital microscopy images. Bar, 50 μm. **f** Quantitative analysis of platelet recruitment by measuring mean fluorescence intensity (MFI) at the site of injury versus background fluorescence ($n = 4$). **g, h** Hemostatic parameters. **g** Tail bleeding time ($n = 4$). **h** Clotting time and clot formation time induced by either extrinsic or intrinsic activation of coagulation ($n = 6$). Graphs show mean and SEM. All $P$-values were determined by unpaired $t$-test except for h (extrinsic clotting time, Mann–Whitney $U$-test)

release their nuclear material in the form of decondensated nucleosomes (neutrophil extracellular traps, NETs) to induce platelet activation and initiate coagulation[16,17]. Neutrophils also release their granule content containing various enzymes that promote blood coagulation[14]. The role of innate immune cells in large-vessel thrombosis and specifically the molecular cues linking leukocytes with thrombus formation are, however, incompletely understood. Here we examine the role of cathelicidins in thrombotic and inflammatory conditions.

Cathelicidins are antimicrobial peptides that form an integral effector component of the innate immune system in vertebrates. The only human cathelicidin identified to date is LL-37, which derives from its precursor protein hCAP18 through proteolytic cleavage[18]. The respective homologue in mice is cathelicidin-related antimicrobial peptide (CRAMP). LL-37 was detected in neutrophils and other leukocyte populations (i.e., lymphocytes, monocytes, and eosinophils)[19–22]. It is also present in tissues, where high local concentrations can be reached[23]. LL-37/CRAMP exerts broad antimicrobial effects in response to bacterial[24] and viral infections[25] and, dependent on the context and leukocyte subset involved, both pro- and anti-inflammatory effects of this peptide have been described[26]. These complex and differential functions were assigned to the ability of LL-37 to activate a broad variety of receptors including the formyl peptide receptor (FPR2),

chemokine (C-X-C motif) receptor 2 (CXCR2), or purinergic receptors (e.g., P2X7 ionotropic receptor)[18]. LL-37/CRAMP induces a proinflammatory phenotype in endothelial cells[27], and absence of CRAMP is associated with reduced atherosclerosis[28,29]. However, while LL-37/CRAMP has been extensively studied in classical inflammatory cells, the effects of cathelicidins on platelets remain elusive. Here, we show that cathelicidins directly activate platelets and foster platelet–neutrophil interactions. Deletion of hematopoietic CRAMP reduces arterial thrombus formation and abrogates inflammation-induced pulmonary injury. Thus, we identify CRAMP/LL-37 as an important mediator of thrombo-inflammation.

## Results
**Cathelicidins are present in human and mouse arterial thrombi.** Recent histological studies in human patients with AMI showed that accumulation of immune cells, and specifically neutrophils, represents a hallmark of coronary artery thrombosis[12,13]. The role of immune cell-derived molecules in thrombosis is, however, incompletely understood. We analyzed tissue taken from five patients with AMI and found that cathelicidins were abundant in arterial thrombi (Fig. 1a, b and Supplementary Fig. 1a, b). LL-37 was not only concentrated within leukocytes

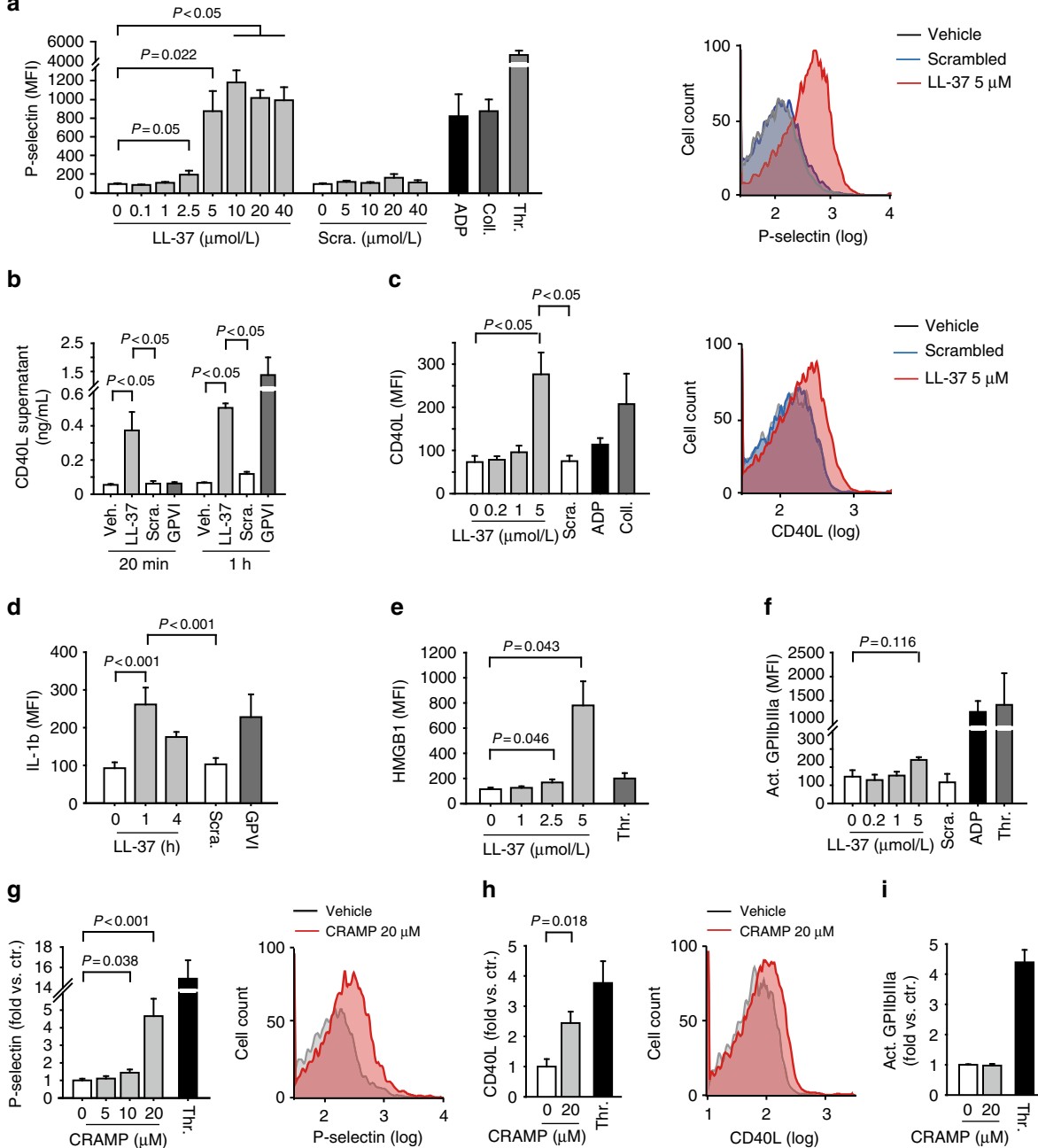

**Fig. 3** Cathelicidins induce platelet activation and secretion. **a–f** Incubation of isolated human platelets for 15 min with increasing concentrations of LL-37 or scrambled control peptide (Scra; applied Scra concentration as indicated or equivalent to the maximal LL-37 concentration). ADP (5 µmol/L), collagen (Coll, 5 µg/mL), thrombin (Thr, Thrombin 0.05 U/mL), or GPVI receptor activating antibody HGP4C9 (1 µg/mL) were used for comparison. **a** Flow cytometric analysis of P-selectin surface expression ($n = 6$). **b** CD40L release into the supernatant as measured by ELISA ($n = 3$). Flow cytometry analysis of **c** CD40 ligand (CD40L) surface expression ($n = 4$), **d** intracellular IL-1β ($n = 5$), **e** HMGB1 surface expression ($n = 4$), and **f** GPIIb/IIIa receptor activation ($n = 6$). **g–i** Flow cytometry analysis of CRAMP-induced activation of isolated mouse platelets. **g** P-selectin ($n = 5$), **h** CD40L ($n = 4$), and **i** activated GPIIb/IIIa ($n = 7$). Thrombin (Thr, 0.5 U/mL) was used for comparison. Graphs show mean and SEM. $P$-values were determined by one-way repeated measures ANOVA with Bonferroni correction (**a**, **c–e**), paired $t$-test (**b**, **f**), or unpaired $t$-test (**g–i**)

(Fig. 1a, b and Supplementary Fig. 1a, c), but also associated with areas of the platelet-rich thrombus in which leukocytes were mostly absent (Fig. 1a). In mice, CRAMP was present in arterial thrombi induced by injury of the carotid artery. Similar to human thrombi, CRAMP was readily detectable in neutrophils but was also found in leukocyte-poor areas of the thrombus (Fig. 1c, d and Supplementary Fig. 2a), suggesting that cathelicidins associate with platelets. To further investigate this interaction, we injected fluorescent CRAMP into wild type (WT) mice and induced injury

of the carotid artery. Using intravital microscopy, we observed that CRAMP-5-FAM, but not the scrambled 5-FAM-control peptide, associated with platelets at the site of platelet-thrombus formation (Fig. 1e, Supplementary Movie 1, 2). Fluorescent cathelicidin also bound to isolated platelets in vitro as determined by flow cytometry (Fig. 1f). In summary, the experiments indicate that LL-37/CRAMP binds to platelets. We were therefore interested in studying the role of cathelicidins for platelet function and thrombus formation.

Together with platelets, neutrophils were rapidly recruited to the injured carotid artery (Supplementary Fig. 3a). Depletion of neutrophils before arterial injury abrogated the accumulation of CRAMP in mouse thrombi, indicating that neutrophils were the most relevant source of cathelicidins in the blood circulation (Supplementary Fig. 3b). While cathelicidins were abundant in arterial thrombi, CRAMP plasma levels did not change significantly in response to arterial thrombosis (Supplementary Fig. 3c), suggesting a local enrichment within the forming thrombus.

**Cathelicidins contribute to arterial thrombosis.** In order to investigate the role of hematopoietic cathelicidins in thrombo-inflammation, we generated chimeric mice by transplanting bone marrow (BM) of WT (wt → wt) or $Cramp^{-/-}$ (ko → wt) mice.

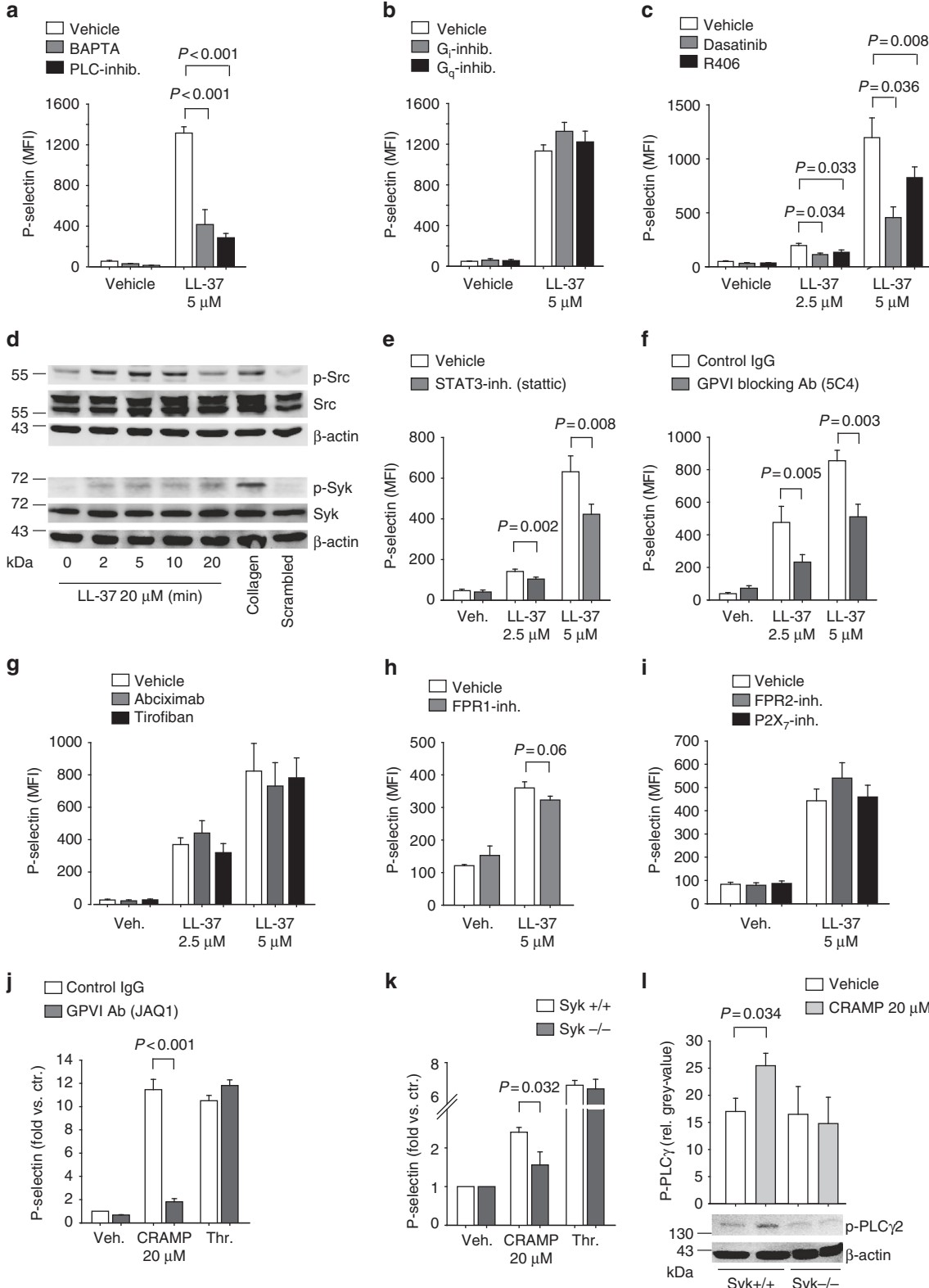

Absence of CRAMP led to reduced thrombus formation (Fig. 2a and Supplementary Movie 3, 4). Time to complete vessel occlusion was prolonged (Fig. 2b), maximal thrombus size reduced (Fig. 2c), and the duration of vessel occlusion was shortened (Fig. 2d), suggesting that hematopoietic CRAMP contributes to arterial thrombus formation and stability. To address the role of CRAMP in platelet accumulation at the site of injury, we carried out ligation of the carotid artery, which produces a more localized injury than that induced by ferric chloride (FeCl₃). Carotid artery ligation, resulted in rapid platelet adhesion and aggregate formation, which was diminished in CRAMP-deficient BM chimera (Fig. 2e, f). Taken together, these data show that absence of CRAMP reduces platelet activation and arterial thrombosis. Bleeding time and plasma clotting time were not altered in CRAMP-deficient BM chimera (Fig. 2g, h). Likewise, blood cell counts and baseline platelet activation markers were similar to controls (Supplementary Table 1, 2).

Neutrophil activation was a common feature in arterial thrombosis. Approximately 60% of Ly6G+ cells stained positive for citrullinated histone 3 (citH3), a marker for neutrophil priming toward NETosis[30–32]. Similar numbers of citH3+ neutrophils were found in thrombi of WT and $Cramp^{-/-}$ BM chimeric mice (Supplementary Fig. 4a-d).

**Cathelicidins induce platelet activation and secretion**. The impact of CRAMP on arterial thrombosis and on platelet aggregation at the arterial wall (Fig. 1), led us to speculate of a role for cathelicidins in platelet activation. To identify the underlying mechanisms we carried out in vitro experiments using isolated human and mouse platelets. LL-37 dose-dependently induced alpha degranulation of human platelets as shown by P-selectin (Fig. 3a, Supplementary Fig. 5a) and CD40L surface expression, as well as by CD40L release into the supernatant (Fig. 3b, c). In addition, LL-37 induced expression of the inflammatory molecules IL-1β (Fig. 3d) and HMGB1 (Fig. 3e). These effects were not observed upon incubation with a scrambled peptide, which served as negative control. In contrast, stimulation of isolated platelets with LL-37 had no effect on activation of the integrin GPIIb/IIIa (Fig. 3f). Furthermore, LL-37 did not induce platelet spreading on fibrinogen-coated slides (Supplementary Fig. 6a). Consequently, LL-37 does not induce platelet aggregation itself (Supplementary Fig. 6b), nor does it influence ADP-, Collagen-, or thrombin-receptor activating peptide (TRAP)-induced aggregation in citrate- or heparin-anticoagulated platelet rich plasma (PRP) (Supplementary Fig. 6c).

To determine whether the mouse homologue CRAMP exerts effects on mouse platelets, we isolated platelets from WT C57Bl/6 mice and stimulated them in vitro with increasing concentrations of CRAMP. Similar to the effects observed in human platelets, stimulation of mouse platelets with CRAMP-induced upregulation of platelet P-selectin (Fig. 3g) and CD40L surface expression (Fig. 3h), but had no effect on integrin GPIIb/IIIa activation as measured by JON-A-binding (Fig. 3i).

**Cathelicidin-dependent signaling in platelets**. We next analyzed the signaling pathways underlying cathelicidin-induced activation of human and mouse platelets using different inhibitors. Blocking of intracellular calcium release by BAPTA (15 μmol/L) and of phospholipase C (U-73122 5 μmol/L) reduced P-selectin and CD40L surface expression of human platelets in response to LL-37 (Fig. 4a and Supplementary Fig. 7a). While blocking of G-protein signaling (pertussis toxin 100 ng/mL or cholera toxin 5 ng/mL) had no effect on LL-37 induced platelet activation (Fig. 4b and Supplementary Fig. 7b), prior inhibition of the tyrosine kinases Syk (R406, 5 μmol/L) and Src-family kinases (Dasatinib, 1 μmol/L) reduced LL-37 induced platelet activation (Fig. 4c and Supplementary Fig. 7c). Accordingly, LL-37 led to phosphorylation of tyrosine kinases Syk and Src (Fig. 4d, Supplementary Fig. 8a, b). Furthermore, inhibition of STAT3 (Stattic 20 μmol/L), which has been attributed non-transcriptional functions in tyrosine kinase signaling in platelets[33], also attenuated LL-37 induced platelet activation (Fig. 4e). Scavenging of extracellular calcium (EGTA 2 mmol/L) or blocking of calpain (MG101 20 μmol/L), phosphoinositide 3-kinase (wortmannin 1 μmol/L), protein kinase C (GF109203X 10 μmol/L), or p38-MAPK (SB203580 10 μmol/L), had no effect on LL-37-induced P-selectin surface expression on platelets (Supplementary Fig. 7d, e). Similar effects were observed for LL-37-induced surface expression of CD40L (Supplementary Fig. 7f, g).

We next addressed the surface receptors involved in LL-37 mediated platelet activation. Addition of a GPVI receptor blocking antibody (HGP5C4) decreased LL-37 induced activation of isolated human platelets (Fig. 4f). Inhibition of GPIIb/IIIa-receptors using Abciximab (10 μg/mL) or Tirofiban (1 mg/mL, Fig. 4g), as well as inhibition of FPR1 (Boc-MLF 10 μmol/L), FPR2 (WRW4 1 μmol/L), or purinergic P2X7 receptor (A438079, 1 μmol/L) did not influence LL-37-induced P-selectin expression on platelets (Fig. 4h, i). Comparable results were also observed for LL-37-induced surface expression of CD40L (Supplementary Fig. 7h-j).

Similar to man, CRAMP-induced upregulation of P-selectin on mouse platelets was inhibited in the presence of a GPVI receptor-depleting antibody (JAQ1, Fig. 4j). Furthermore, CRAMP-induced platelet activation was decreased in $PF4^{Cre}:Syk^{flox/flox}$ mice with platelet deficiency of Syk (Fig. 4k). Platelets of these animals also showed no increase in phosphorylation of PLCγ compared with respective controls (Fig. 4l, Supplementary Fig. 8c). These data infer that both human and mouse cathelicidins elicit platelet activation involving, at least in part, GPVI receptor and calcium-dependent downstream signaling via Syk and PLC.

**Cathelicidins induce platelet–neutrophil interactions**. Cathelicidins induced robust platelet activation with P-selectin and HMGB1 surface expression (Fig. 2). We therefore investigated whether this pathway promotes platelet–neutrophil interactions. Using flow cytometric analysis we found that platelet pre-

**Fig. 4** Cathelicidin-dependent signaling in platelets. **a–i** LL-37 induced signaling in isolated human platelets. **a–c** Flow cytometry analysis of platelet P-selectin surface expression in the presence of **a** the calcium chelator BAPTA (n = 5) or a phospholipase C inhibitor (U-73122, n = 6), **b** pertussis or cholera toxin to inhibit G-protein signaling (n = 4), or **c** inhibitors of tyrosine kinases Src-family kinases (Dasatinib, n = 7) and Syk (R406, n = 5). **d** Representative western blots of phosphorylated Src-family kinase and phosphorylated Syk upon incubation of platelets with LL-37. Collagen was used as positive control for tyrosine kinase phosphorylation, β-actin served as loading control. Images are representative of three independent blots. **e–i** Flow cytometry analysis of LL-37 platelet P-selectin surface expression in the presence of **e** STAT3 small molecule inhibitor (Stattic, n = 6), **f** GPVI antibody (HGP5C4, n = 9), **g** GPIIb/IIIa antibody (Abciximab or Tirofiban, n = 4), **h** formyl-peptid-receptor (FPR1 or FPR2) antibody, and **i** inhibitors against the purinergic P2X₇-receptor (Boc-MLF 10, WRW4 or A438079, n = 4). **j–l** CRAMP-induced signaling in isolated mouse platelets. **j** P-selectin surface expression in the presence of a GPVI depleting antibody (JAQ1, n = 6). **k** P-selectin surface expression and **l** phospholipase C phosphorylation in platelet lysates from platelet-specific Syk-deficient mice and respective littermates after stimulation with CRAMP (flow cytometry: n = 10 $Syk^{-/-}$ and n = 5 $Syk^{+/+}$ animals, western blot analysis n = 5 each). Graphs show mean and SEM. P-values were determined by unpaired (**a**, **f**, **j–l**) or paired (**c**, **e**, **h**) t-test

stimulation with LL-37 increased platelet–neutrophil aggregates formation as compared with platelets incubated with a scrambled version of the peptide (Fig. 5a, Supplementary Fig. 5b). Formation of platelet–neutrophil aggregates was prevented by a blocking antibody against P-selectin (Fig. 5b). These interactions were accompanied by neutrophil activation, as indicated by increased CD11b expression (Fig. 5c), production of reactive oxygen species

(ROS) (Fig. 5d) and L-selectin shedding (Fig. 5e) after incubation with LL-37 stimulated platelets. LL-37 pre-stimulated platelets also induced the release of extracellular nucleosomes from neutrophils (Fig. 5f). In contrast, NET formation was not observed in the absence of platelets or following pre-stimulation with the scrambled peptide (Fig. 5g, h). Both neutrophil activation (Fig. 5c) and NET formation (Fig. 5g, h) was abrogated in the

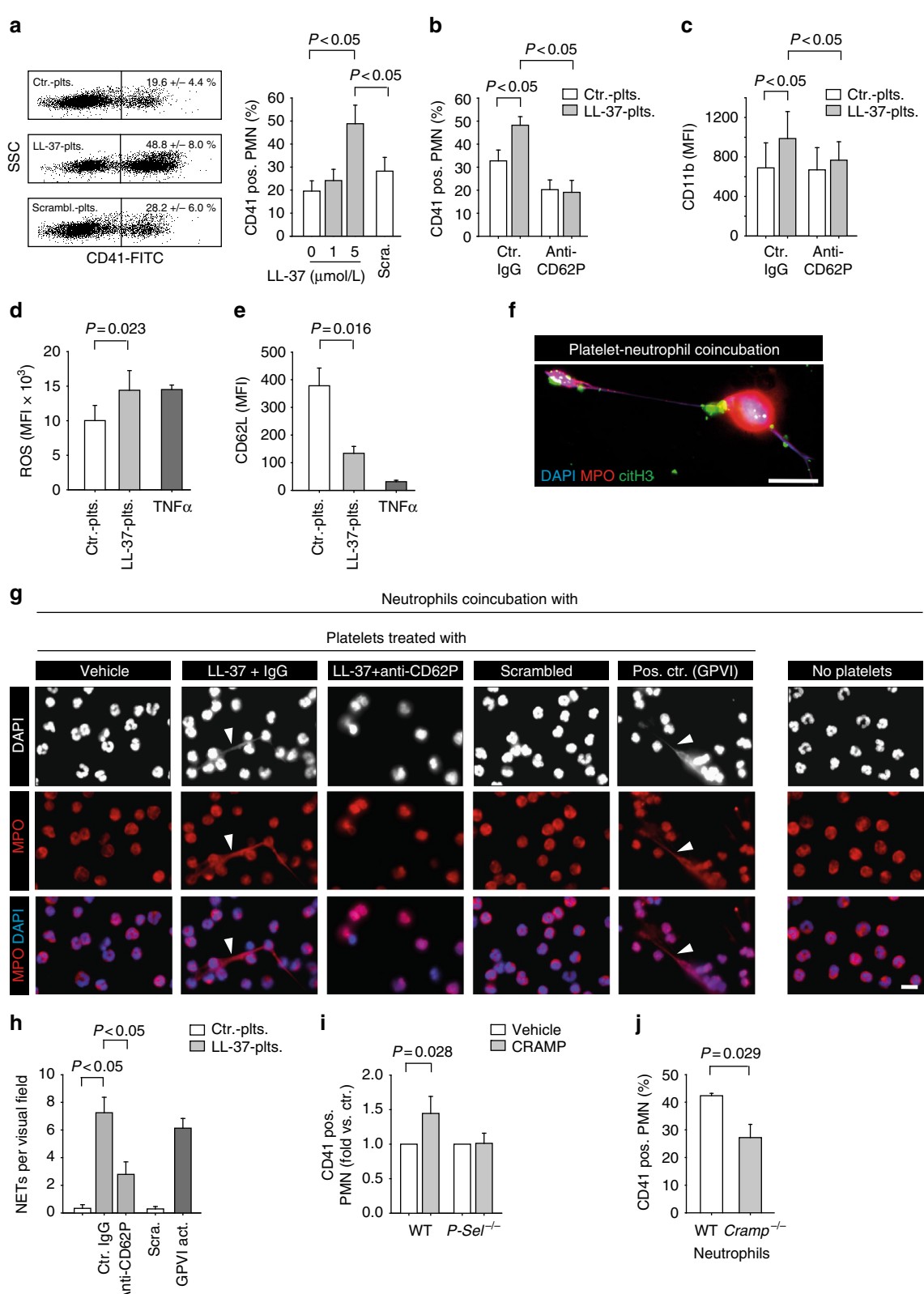

**Fig. 5** Cathelicidins induce platelet–neutrophil interactions. **a–h** Co-incubation experiments. Human platelets were pretreated with LL-37 or scrambled control peptide (Scra) and platelet–neutrophil interactions were analyzed. **a–e** Flow cytometry analysis of **a** platelet–neutrophil aggregates formation ($n = 9$), **b** platelet–neutrophil aggregates in the presence of a blocking antibody against P-selectin and respective isotype control ($n = 5$), **c** CD11b expression on neutrophils, **d** neutrophil intracellular formation of reactive oxygen species (ROS), **e** shedding of neutrophil L-selectin ($n = 4$). TNF$\alpha$ (50 ng/mL) served as positive control. **f–h** Neutrophil extracellular trap (NET) formation assay. **f** Representative epifluorescence image of a NET. DAPI (nuclear stain, blue), myeloperoxidase (MPO, red), and citrullinated histone H3 (citH3, green). Bar, 10 μm. **g** NET formation was induced by platelets that were pretreated with LL-37 or a GPVI-activating antibody (HGP4C9). Upper row (DAPI nuclear stain, white), middle row (MPO, red), and bottom row (merged image of DAPI in blue, and MPO in red). Arrowheads indicate NET. Bar, 10 μm. **h** Quantitative analysis of NET formation ($n = 4$). **i, j** Interactions of mouse cells. **i** Platelet–neutrophil aggregates formation of mouse neutrophils with platelets isolated from wild type (WT) or P-selectin deficient mice ($n = 7$). **j** Platelet–neutrophil aggregates formation after co-incubation of isolated WT platelets with PMA (50 μmol/L) activated neutrophils of WT or CRAMP$^{-/-}$ mice ($n = 4$). Graphs show mean and SEM. *P*-values were determined by one-way repeated measures ANOVA with Bonferroni correction (**a–c**), paired *t*-test (**d**, **e**), ANOVA on Ranks/Dunn's method (**h**) or Mann–Whitney *U*-test (**i**, **j**)

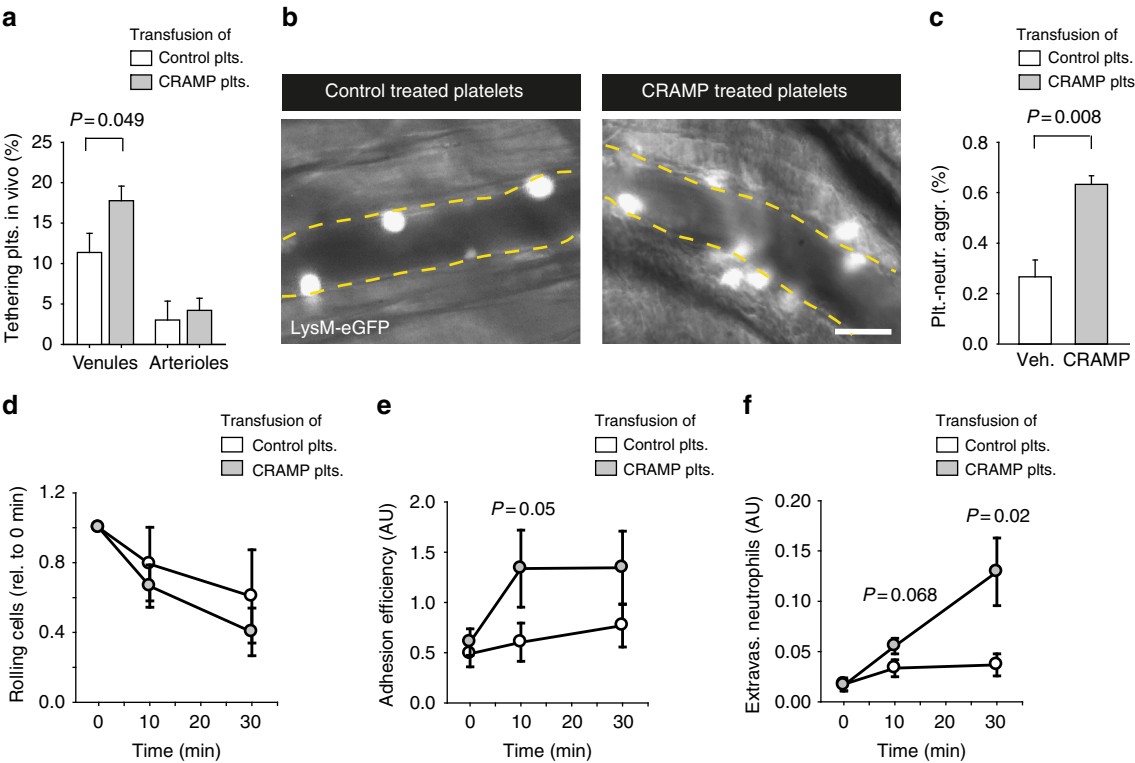

**Fig. 6** CRAMP-activated platelets promote neutrophil recruitment and extravasation. **a–f** Intravital microscopy of the mouse cremaster muscle following trauma-induced inflammation. Platelets were isolated and pretreated ex vivo with CRAMP or vehicle. $10^7$ platelets were then injected into lysozyme 2 (LysM)-eGFP (neutrophil reporter) recipient mice. **a** Tethering of CRAMP-pretreated platelets in cremaster muscle venules and arterioles ($n = 8$ vessels each, see also Supplemental Movies 3, 4). **b** Representative intravital epifluorescence microscopy images of neutrophils (white). Bar, 20 μm. **c** Platelet–neutrophil aggregates formation in whole blood of GFP-positive neutrophils and transfused platelets that were pretreated with CRAMP or vehicle control ($n = 3$). **d** Neutrophil rolling, **e** adhesion efficiency, and **f** extravasation in cremaster muscle venules ($n = 10$–12 vessels of four different animals). Graphs show mean and SEM. *P*-values were determined by unpaired *t*-test

presence of a P-selectin antibody. These data show that cathelicidin-primed platelets can induce neutrophil activation and NET release.

To further analyze the role of CRAMP for platelet–neutrophil interactions in vitro, we performed co-incubation assays of isolated mouse platelets with BM-derived mouse neutrophils. Platelets prestimulated with CRAMP showed increased platelet–neutrophil aggregates formation compared with unstimulated platelets (Fig. 5i). This effect was blunted in co-incubation assays carried out with platelets from P-selectin deficient (*P-Sel*$^{-/-}$) mice (Fig. 5i). Conversely, phorbol ester (50 μmol/L phorbol 12-myristate 13-acetate (PMA)) activated neutrophils from *Cramp*$^{-/-}$ mice had reduced aggregate formation with WT platelets compared with neutrophils from *Cramp*$^{+/+}$ mice (Fig. 5j). Absence of CRAMP did not confer differences in P-selectin, CD40L or activated GPIIb/IIIa expression on platelets under steady-state conditions in vivo

(Supplementary Table 3). In summary, neutrophil-derived cathelicidins elicit platelet activation and platelet–neutrophil interactions in vitro.

**CRAMP-activated platelets promote neutrophil extravasation.** Next, we investigated whether CRAMP-induced platelet activation influences inflammatory processes in vivo. Platelets isolated from WT donor mice, labeled with rhodamine-6G and treated with CRAMP (20 μmol/L) or vehicle were injected into lysozyme M (LysM)-eGFP mice, in which mostly neutrophils but also a fraction of monocytes express the green fluorescent protein (GFP)[34]. Intravital microscopy was carried out in a model of trauma-induced inflammation of the cremaster muscle. CRAMP-pretreatment of donor platelets increased platelet tethering in postcapillary venules (Fig. 6a, b and Supplementary Movie 5, 6).

Moreover, CRAMP-pretreated platelets formed more aggregates with neutrophils as measured ex vivo in whole blood (Fig. 6c). While the number of rolling leukocytes was not affected (Fig. 6d), injection of CRAMP-pretreated platelets enhanced trauma-induced adhesion (Fig. 6e), as well as extravasation (Fig. 6f) of neutrophils in cremaster muscle venules. Thus, cathelicidin-induced platelet–neutrophil interactions promote neutrophil extravasation at sites of tissue inflammation.

**Cathelicidins contribute to ALI**. Acid-induced ALI in mice is driven by platelet–neutrophil interactions[35]. We therefore aimed to determine whether cathelicidins play a role in pulmonary injury and inflammation. LL-37 was abundant in lung specimen of patients with pneumonia (patients 1–4), including a case of aspiration pneumonia (patient 1). LL-37 expression was associated with the infiltration of inflammatory cells (Fig. 7a, b). We then investigated the role of the mouse cathelicidin CRAMP in a

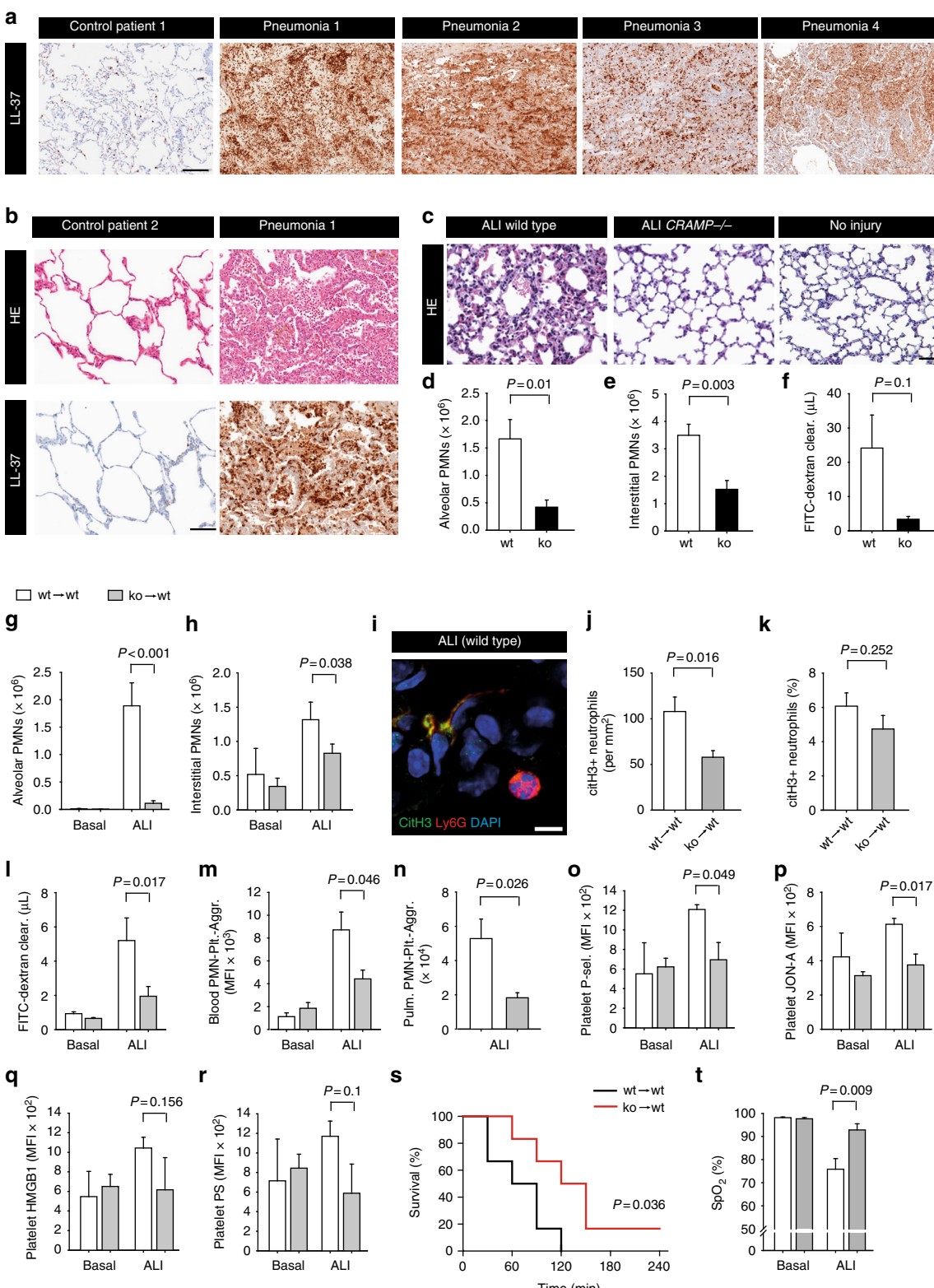

**Fig. 7** Cathelicidins promote lung injury. **a, b** Representative histology images of lung tissue obtained from patients with acute pneumonia and two control subjects without tissue inflammation. Patient 1 presented with aspiration pneumonia. Immunohistochemistry for LL-37 (**a, b**) and hematoxylin-eosin (H&E) staining (**b**). Bars, 100 μm (**a**) or 50 μm (**b**). **c–t** Mouse model of acute lung injury (ALI) induced by intratracheal injection of 0.1 M HCl. **c** Representative images of H&E stained mouse lung tissue after ALI in wild type and CRAMP$^{-/-}$ mice. Bar, 50 μm. **d** Alveolar neutrophil counts, **e** number of interstitial (including pulmonary intravascular) neutrophils, and **f** alveolar permeability after ALI in wild type ($n = 6$) and CRAMP$^{-/-}$ mice ($n = 7$). **g** Alveolar neutrophils and **h** interstitial (including pulmonary intravascular) neutrophils after ALI in wild type (wt → wt) and CRAMP$^{-/-}$ (ko → wt) chimeras ($n = 7$). **i–k** Analysis of citrullinated histone H3 (citH3) staining in Ly6G+ neutrophils. **i** Representative image of lung tissue after ALI. CitH3 (green), Ly6G (red), DAPI (nuclear stain, blue). Bar, 10 μm. **j, k** Quantification of citH3 staining in Ly6G+ neutrophils ($n = 6$). **l** Alveolar permeability after ALI ($n = 7$). **m, n** Flow cytometry analysis of platelet–neutrophil aggregates after ALI in **m** systemic circulation ($n = 6$) and **n** the pulmonary vascular compartment ($n = 7$). **o–r** Flow cytometry analysis of **o** P-selectin surface expression, **p** GPIIb/IIIa activation, **q** HMGB1 surface expression, and **r** phosphatidylserine (PS) exposure on platelets. **s** Survival curves and **t** oxygen saturations after 30 min after ALI induction ($n = 6$). Graphs show mean and SEM. P-values were determined by Mann–Whitney U-test (**d, f, g, h, l, n**), unpaired t-test (**e, j, k, m, o, p, q, r, t**), or Log-rank (Mantel-Cox) test (**s**)

mouse ALI model in vivo. *Cramp*$^{-/-}$ mice compared with WT animals showed lower amounts of neutrophils in the alveolar space (Fig. 7c, d, Supplementary Fig. 5c), as well as in the interstitial (including pulmonary vascular) compartment (Fig. 7e). To define the contribution of hematopoietic CRAMP to these effects, we carried out the ALI model in BM chimeric mice. CRAMP-deficient BM chimeric mice displayed reduced amounts of neutrophils in the alveolar space (Fig. 7g) and in the interstitial (including pulmonary intravascular) compartment (Fig. 7h). citH3+ neutrophils were also detected in lung tissue after ALI, albeit with lower frequency (5–6%) as compared to arterial thrombi (Fig. 7i). Absence of CRAMP reduced the absolute number of citH3+ neutrophils (Fig. 7j), but did not alter the percentage of citH3+ cells in the ALI model (Fig. 7k).

CRAMP-deficient BM chimeric mice showed decreased alveolar permeability (Fig. 7l). Ex vivo analysis revealed a reduction in platelet–neutrophil aggregates in the systemic circulation and the vascular compartment of the lung in CRAMP-deficient chimeras (Fig. 7m, n). Circulating platelets of CRAMP-deficient chimeras also showed reduced surface expression of P-selectin and decreased activation of GPIIb/IIIa compared with WT BM chimeras (Fig. 7o, p), but reduction in surface HMGB1 and phosphatidylserine did not reach statistical significance (Fig. 7q, r). Importantly, absence of CRAMP in BM chimeric mice was associated with a survival benefit (Fig. 7s) and an improved oxygen saturation following ALI (Fig. 7t).

In summary, cathelicidins are abundant in inflamed lung tissue of mice and men. Absence of hematopoietic CRAMP reduces platelet–neutrophil interactions and abrogates ALI. These findings suggest that cathelicidins are important signaling molecules provided by neutrophils that activate platelets and promote inflammatory conditions and acute thrombosis.

## Discussion

We report here that hematopoietic LL-37/CRAMP induces platelet activation and secretion, which mediates platelet–leukocyte aggregate formation and neutrophil recruitment at sites of inflammation. We identified LL-37/CRAMP in thrombi and demonstrated that depletion of CRAMP reduces the development of arterial thrombosis in mice. Further, in a mouse model driven by platelet–neutrophil interactions, absence of CRAMP abrogated pulmonary injury. Inhibition of hematopoietic cathelicidin may therefore represent a novel strategy to reduce thrombo-inflammatory disease.

Arterial thrombosis is a common condition with high morbidity and mortality. While platelets are considered key elements of this process, immune cells have recently been identified in coronary artery thrombi of patients with AMI[12,13]. The implications of these findings are only partly understood, but the local release of leukocyte content at sites of vascular injury has been implicated in promoting thrombosis. Neutrophil-derived

nucleosomes form a pro-thrombotic matrix that activates platelets and stimulates factor XII-dependent coagulation[14,36]. In addition, enzymes, such as serine proteases, cleave tissue factor pathway inhibitor thereby enhancing TF-dependent coagulation[14]. Immune cell-derived molecules therefore seem to play a role in large-vessel thrombosis. While various non-hematopoietic cells (endothelia and epithelia) release LL-37/CRAMP upon activation[24,37], our findings in *Cramp*$^{-/-}$ BM chimeric mice demonstrate the relevance of hematopoietic CRAMP in arterial thrombosis and lung injury. Furthermore, depletion of neutrophils from the blood circulation diminished CRAMP in arterial thrombi, suggesting that neutrophils are the most relevant source of hematopoietic LL-37/CRAMP in the pathologic conditions analyzed. Together with platelets, neutrophils are rapidly recruited to the site of arterial injury. Through their release of LL-37/CRAMP, neutrophils contribute to the activation and aggregation of platelets, to promote thrombosis. We provide evidence that cathelicidins directly bind to platelets in vitro, and fluorescence-tagged CRAMP, but not the scrambled version of the cathelicidin peptide, readily associates with arterial thrombi in vivo. While it was not possible to visualize the release of cathelicidins from neutrophil granules and their direct binding to platelets by intravital microscopy due to technical limitations, our experiments clearly demonstrate the enrichment of cathelicidins within arterial thrombi. Further, the immunohistochemistry of both human and mouse arterial thrombi indicates that cathelicidins are not only concentrated within neutrophils but also associate with thrombus areas that are platelet rich and devoid of leukocytes. In mice with genetic absence of CRAMP we then show that thrombus growth and stability is significantly reduced. Taken together, these experiments indicate that neutrophil-derived LL-37/CRAMP accumulates in arterial thrombi and promotes platelet activation and thrombus formation.

The physiological concentrations of cathelicidins prevailing at sites of injury are incompletely understood, which poses a potential limitation. Analysis of bronchoalveolar lavage (BAL) fluid and samples taken from different mucosal sites estimates concentrations to be around 1 and 10 μmol/L for LL-37 and CRAMP, respectively[38–40], which are much higher than the concentrations reached in plasma[23]. The in vitro concentrations applied in this study (5 μmol/L LL-37, and 20 μmol/L CRAMP) are in this order of magnitude. Further, in our histological analysis we found that cathelicidins were abundant in inflamed lungs and in arterial thrombi, which is in line with the local enrichment of LL-37/CRAMP within other tissues[23,38–40]. However, measuring in vivo levels of cationic peptides has been challenging[41], and the precise concentrations reached locally at the site of tissue injury remain to be determined. In the concentrations indicated above, LL-37/CRAMP induced activation of isolated human and mouse platelets resulting in the secretion of the adhesive molecules P-selectin and CD40L as well as other inflammatory molecules such as IL-1β and HMGB1. In vitro, LL-37 did not

influence agonist-induced platelet aggregation, which is in contrast to effects described at very high and potentially artificial concentrations (0.1–1 mM)[42], at which cathelicidins exert cytotoxic effects[43].

In immune and cancer cells, LL-37 has been shown to activate various intracellular signaling pathways, including calcium-dependent activation of PI3K/Akt and PKC/MAP kinases[18]. These signaling cascades have a role in both classical (aggregation) and nonclassical (immunity) platelet functions. We found that LL-37-mediated platelet activation is, at least in part, calcium dependent involving signaling via Syk. Notably, platelet activation via the non-receptor protein tyrosine kinase Syk has been associated with thrombo-inflammatory conditions, such as autoimmune thrombocytopenia and cerebral infarction[44], and inhibition of Syk may yield novel treatment options therein[45,46].

In platelets, Syk signaling is associated with receptors that provide an immunoreceptor tyrosine-based activation motif, such as the collagen receptor complex GPVI-FcR γ-chain[47]. Application of GPVI antibodies inhibited cathelicidin-induced platelet activation indicating that this pathway plays a mechanistic role. GPVI is the central platelet collagen receptor, which mediates platelet adhesion and thrombosis following vascular injury[48]. Recent work has linked GPVI to immune and inflammatory processes[49,50]. Notably, ligation of the GPVI receptor has been shown to promote proinflammatory actions of platelets, such as neutrophil activation and secretion, which is an important mechanism in tissue injury[7,50]. However, other receptors may have a role in this context, specifically in humans, where GPVI-blockade only partly inhibited platelet activation by LL-37. The C-type lectin CLEC-2 could represent such an alternative receptor, as this has also been shown to signal through Syk[51]. In contrast, neither pharmacological inhibition of G-protein coupled receptors, FPRs nor the purinergic P2X$_7$ receptor, which modulate immune cell functions and chemotaxis[52], altered LL-37 induced platelet activation at indicated concentrations in vitro.

In addition to arterial thrombosis, we identified cathelicidin-induced platelet–neutrophil interactions to play a role in inflammatory scenarios. CRAMP-activated platelets enhanced aggregate formation with neutrophils and fostered their recruitment and extravasation at sites of tissue injury in vivo. Absence of CRAMP-dependent platelet activation reduced cremaster muscle inflammation and pulmonary injury. Because activated platelets stimulate neutrophils and induce their secretion, neutrophil-dependent platelet activation, via LL-37, reinforces bidirectional interactions between these cells (Supplementary Fig. 9). This notion is further supported by our in vitro findings showing that cathelicidin-activated platelets elicit neutrophil activation in various ways. This includes upregulation of CD11b, ROS production, and NET formation. All of these processes could differentially contribute to thrombo-inflammation[53–56]. Whether LL-37 can induce NETosis autonomously in vivo is unclear to date[57]. Our findings, however, suggest that cathelicidins are not required for priming of neutrophils toward NETosis under the conditions analyzed. Notwithstanding the above, cathelicidins can associate with extracellular DNA released by neutrophils[58], and could potentially be presented to nearby platelets to foster thrombo-inflammation.

It has been shown that neutrophils engage activated platelets and recruit them to sites of inflammation to modulate local inflammatory responses[7]. Depletion of platelets, as well as inhibition of surface receptors relevant for platelet–neutrophil interactions, such as P-selectin glycoprotein ligand-1, improves outcome in mouse thrombo-inflammation models, including arterial thrombosis and ALI[7,59]. This is in line with previous work indicating the essential role of platelet–neutrophil interactions in ALI[9]. Given the importance of the platelet–neutrophil interplay

in a number of diseases, a deeper understanding of the exact mechanisms modulating these processes could open new therapeutic avenues. First clinical trials targeting platelet–leukocyte interactions (via P-selectin) to prevent and treat atherosclerosis have recently been initiated in man[60]. Cathelicidins may provide an alternative target in inflammatory conditions and thrombosis.

## Methods

**Human blood and tissue samples**. The study conformed to the principles outlined in the Declaration of Helsinki. All individuals had given their written informed consent, and tissue samples were pseudonymized. Studies on human samples were approved by the Ethics Committee of the University of Munich. Human coronary artery thrombi were obtained from patients with myocardial infarction undergoing catheter thrombectomy (ProntoTM thrombectomy catheter device, Vascular Solutions, Minneapolis, USA) during percutaneous coronary intervention. Samples were immediately submerged in liquid nitrogen after retrieval and stored at −80°C. Paraffin-embedded lung tissue specimens from patients with or without acute pneumonia were drawn from the archives of the Institute of Pathology for studies on human lung tissues. Specimens were anonymized.

**Animal experiments**. All animal procedures were in accordance with the German animal protection law and the Directive 2010/63/EU of the European Parliament and were approved by the Government of Bavaria (Regierungspräsidium Oberbayern), Munich, Germany. Surgical procedures were carried out under short-term anesthesia using midazolam (5 mg/kg), fentanyl (0.05 mg/kg), and medetomidine hydrochloride (0.5 mg/kg). In vivo experiments were carried out using male animals, for ex vivo experiments both males and females were used. Experiments were carried out using 12–20 weeks old mice. Specifically, all mice undergoing BM transplantation were 20 weeks old at the time of final experiments (thrombus formation in vivo, lung injury model) were carried out. All other mice were 12 weeks old at the time of final experiments. All controls were age- and sex-matched. All mice were on C57Bl/6 background. WT mice were obtained from Janvier Labs (France). *Cramp*$^{−/−}$, *P-selectin*$^{−/−}$, *PF4*$^{Cre/+}$*Syk*$^{fl/fl}$, and LysM-GFP mice were described previously[3,29,61,62]. Mice were maintained in a specific pathogen-free environment and fed standard mouse diet ad libitum.

**Generation of *Cramp*$^{−/−}$ BM chimeric mice**. Whole BM cells were isolated from *Cramp*$^{−/−}$ donors and 10$^7$ cells were injected into the tail vein of irradiated recipient mice (2 × 650 Rad with an interval of 8 h, injection was 3 h after second irradiation) to generate of BM chimeras. Experiments were carried out 8–12 weeks after transplantation. Efficiency of chimerism was analyzed 8 weeks after transplantation by flow cytometry and was at least 90%.

**Immunohistochemical staining**. For immunohistochemistry, 5 μm sections of paraffin-embedded tissue samples were deparaffinized, incubated with rabbit anti-human LL-37 antibody (Innovagen, Sweden, Cat# PA-LL-37-100) or rabbit anti-mouse CRAMP antibody (Innovagen, Sweden, Cat# PA-CRPL-100) at 1:300, and stained on a Ventana Benchmark XT autostainer with an ultraView Universal DAB detection kit (Ventana Medical Systems).

**Immunofluorescence analysis**. Frozen tissue samples were cut with a cryotome (CryoStar NX70, ThermoFisher Scientific) into 10 μm sections, fixed with 4% formaldehyde and blocked with the respective serum. The sections were incubated with primary antibodies for 1 h at room temperature. Rabbit polyclonal anti-LL-37 (Cat No. PA-LL-37-100) and rabbit polyclonal anti-CRAMP (Cat No. PA-CRPL-100) antibodies were from Innovagen (Lund, Sweden), mouse monoclonal anti-human myeloperoxidase (MPO) antibody (clone 2C7) was from Abcam (Cat No. ab25989), goat polyclonal anti-human MPO antibody was from R&D Systems (Cat No. AF3174), mouse monoclonal anti-human CD41 (clone HIP8) was from Biolegend (Cat No. 303710), and rat anti-mouse CD41 (clone MWReg30) antibody was from eBioscience (Cat No. 14-0411-82). Mouse monoclonal anti-human histone H3 citrulline antibody (clone 7C10) was from Covalab (Cat No. mab0072-P) and rabbit polyclonal anti-mouse histone H3 (citrulline R2+R8+R17) antibody was from Abcam (Cat No. ab5103). Respective Alexa488-, Alexa555-, and Alexa594-conjugated secondary antibodies were from Invitrogen. DNA was stained with 1 μg/mL Hoechst 33342 (Invitrogen, Cat No. H1399) or 1 μg/mL DAPI (Sigma, Cat No. 28718-90-3), and a coverslip was placed using mounting medium (DAKO, Cat No. S3023). Primary antibodies were applied 1:1000 (final dilution), secondary antibodies and DAPI/Hoechst 1:2000, respectively. Images were acquired using either a Zeiss Imager M2 Axio epifluorescence microscope and processed using AxioVision SE64 Rel. 4.9 software, or a Leica DMRB epifluorescence microscope equipped with a Zeiss AxioCam and processed using AxioVision 4.6 software (Zeiss), or a LSM 880 confocal microscope with Airyscan module and Plan-Apochromat ×20/0.8 air objective (Carl Zeiss Microscopy) and processed using ZEN software (Zeiss).

For cathelicidin staining in mice, spleen of *Cramp*$^{-/-}$ mice served as control (Supplementary Fig. 2b).

**FeCl$_3$-induced thrombus formation in the mouse carotid artery**. The mice were anesthetized using 2% isoflurane and intraperitoneal injection of fentanyl (0.05 mg/kg), midazolam (5.0 mg/kg), and medetomidine (0.5 mg/kg). Thereafter, the right common carotid artery was exposed. FeCl$_3$ (1 µL of 10% FeCl$_3$ soaked in 1 mm$^2$ Whatman paper) was topically applied for 4 min to the common carotid artery. The artery was then thoroughly rinsed with saline and kept moist for the time of observation. For immunohistochemistry or immunofluorescence analysis thrombi were collected surgically 30 min after injury and were immediately embedded in OCT and frozen at −80 °C, or fixed in 4% formaldehyde and embedded in paraffin.

For intravital microscopy, the fluorescent dye DiOC6 (5 µL/g of body weight of a solution with 100 µmol/L) was injected via tail vein before application of FeCl$_3$ to allow visualization and thrombus formation was monitored for 60 min by placing the carotid artery under a fluorescence microscope equipped with a camera (AxioScope; Carl Zeiss). Fluorescent images were acquired sequentially (1 image/s) and thrombus size and kinetics (i.e., time to occlusion, duration of occlusion) were analyzed using AxioVision 4.7 imaging software (Zeiss). In some experiments, neutrophil depletion was performed before thrombus formation using a Ly6G-specific mAb (clone 1A8, BD Biosciences, Cat No. 551459). The antibody was injected intravenously at a concentration of 5 µg/g body weight (in 150 µL PBS) 24 and 6 h before induction of thrombosis[63].

To analyze binding of CRAMP to thrombus in vivo, 5-carboxyfluorescin (5-FAM)-labeled CRAMP peptide (Innovagen, Sweden, CAT No. SP-5352-1) or 5-FAM-labeled scrambled peptide (custom made from Innovagen, Sweden) were infused intravenously (4 µg per gram body weight in 150 µL of sterile PBS) shortly before FeCl$_3$ application. Platelets were labeled by injection of a fluorescently labeled non-blocking platelet antibody (anti-mouse GPIb-DyLight649, Emfret, Cat No. X649) at a concentration of 0.1 µg (1 µL) per gram body weight in 150 µL of sterile PBS. Imaging was performed by videofluorescence microscopy (microscope: Leica DM 6 FS; camera system: Andor Zyla sCMOS).

**Ligation injury of the mouse carotid artery**. Platelet recruitment and aggregate formation was investigated after carotid artery ligation[14]. Vascular injury was induced by ligating the carotid artery with a suture and maintaining obstruction of blood flow for 5 min. Afterwards blood flow was reestablished by removal of the suture. In order to visualize platelet aggregate formation a fluorescently labeled platelet antibody (anti-mouse GPIb-DyLight488, Emfret, Cat No. X488) was injected via the tail vein. Measurements were carried out with a high-speed widefield Olympus BX51WI fluorescence microscope using a long-distance condenser and a ×20 (NA 0.95) water immersion objective with an Olympus MT 20 monochromator and an ORCA-ER CCD camera (Hamamatsu). Cell^R software (Olympus) was used for image recording and analysis.

**Blood cell count**. Differential blood cell counts in mouse blood were performed using an Idexx Procyte Dx hematology analyzer (Idexx Europe, Hoofddorp, the Netherlands).

**Bleeding time**. To assess bleeding time a 5 mm segment of the tail of anesthetized mice was removed with a razor blade. The tail was immediately immersed in 0.9% isotonic saline at 37°C, and the time required to stop spontaneous bleeding was measured.

**Assessment of plasmatic coagulation**. Parameters of the plasmatic coagulation were assessed by thrombelastometry (ROTEM, Tem International GmbH, Germany) according to the instructions of the manufacturer[64]. Clotting time (time to onset of clot formation) and clot formation time (time from onset of clot formation to a clot firmness of 20 mm) were analyzed for the extrinsic, as well as for the intrinsic activation.

**Plasma levels of CRAMP**. Mouse plasma was obtained by centrifugation (2000×g, 5 min) of citrated whole blood taken by cardiac puncture 2 h after FeCl$_3$ induced carotid artery injury. CRAMP levels were measured using an ELISA kit from MyBiosource (San Diego, USA) according to the instructions of the manufacturer.

**Platelet isolation**. For all in vitro blood cell studies, samples were obtained from healthy individuals who had not taken medications for at least 10 days. Human PRP was obtained by centrifugation of anticoagulated (3.13% sodium citrate) whole blood at 340×g for 15 min. After another centrifugation step at 400×g for 10 min, in the presence of 0.5 µg/mL prostaglandin (Sigma), platelets were washed and resuspended in modified Tyrode's solution (138 mmol/L NaCl, 2.7 mmol/L KCl, 12 mmol/L NaHCO$_3$, 400 mmol/L Na$_2$HPO$_4$, 1 mmol/L MgCl$_2$, 5 mmol/L D-glucose, and 5 mmol/L HEPES). Platelet concentration was adjusted to that required for the respective experiment. Platelet counts were obtained using a resistance particle counter (Coulter Z2, Beckman, Krefeld, Germany).

Isolation of mouse platelets followed the same protocol but with minor modifications. Blood was drawn from the inferior vena cava of anesthetized mice.

Centrifugation steps were 130×g for 5 min for preparation of PRP and 340×g for 10 min for isolation of platelets.

**Platelet stimulation**. Antimicrobial peptides (LL-37, CRAMP) and scrambled peptide were obtained from Innovagen (Lund, Sweden). Stimulation of platelets was carried out with the time and concentrations indicated. As positive controls bovine thrombin (Sigma), TRAP (Roche, Switzerland), ADP (Sigma), Horm Collagen (Takeda), or GPVI-activating antibody HGP4C9[65] were used.

**Cathelicidin binding to platelets**. Isolated washed human platelets were incubated with 5-FAM labeled LL-37 (5 µmol/L) or the same concentration of a 5-FAM labeled scrambled peptide (Innovagen, Sweden). After washing, binding to platelets was assessed by flow cytometry on a BD LSR Fortessa (BD Biosciences, USA).

**Flow cytometric analysis of platelet activation**. Resting or stimulated isolated human or mouse platelets were stained with labeled antibodies or respective isotype controls for 15 min at 37 °C. Human platelets were incubated with antibodies detecting P-selectin (BD Biosciences, clone AK4, Cat No. 550888), CD40L (BD Biosciences, clone M90.1, Cat No. 552559), activated GPIIb/IIIa (PAC-1, BD Biosciences, Cat No. 340507) or high-mobility group box 1 protein (R&D Systems, clone #115603, Cat No. IC1690G). Mouse platelets were incubated with antibodies detecting P-selectin (BD Biosciences, clone RB40.34, Cat No. 561923), CD41 (eBioscience, clone MWReg30, Cat No. 14-0411-82), CD61 (BD Pharmingen, clone 2C9.G2, Cat No. 561911), CD42b (Emfret, clone Xia.G5, Cat No. M040-3), GPVI (R&D Systems, clone #784808, Cat No. MAB6758-SP), GPIX (Emfret, clone Xia.B4, Cat No. M051-1), CD40L (BD Biosciences, Clone MR1, Cat No. 553658), activated GPIIb/IIIa (JON-A, Emfret, Cat No. M023-2). In addition, we carried out Annexin-V (BD Pharmingen, Cat No. 556420) staining. P-selectin antibodies were applied 1:20 (final dilution), all other antibodies 1:50. In another set of experiments to assess platelet activation in mice ex vivo, whole blood was incubated with respective antibodies and red cells were lysed with FACS lysing solution (BD Biosciences, Cat No. 349202) before analysis. All samples were analyzed using a FACS Canto II flow cytometer (BD Biosciences).

**Platelet secretion**. Platelet CD40L secretion was measured using a human CD40L ELISA kit (Biozol Diagnostica, Houston, USA). Therefore, supernatant of LL-37 or control treated platelets was acquired by repeated centrifugation. Supernatants were snap frozen and stored at −80 °C. Measurements were carried out according to the manufacturer's instructions. Intracellular levels of IL-1β were measured by flow cytometry analysis. Washed human platelets were stimulated for the indicated times and fixed using 1% PFA, permeabilized using 0.1% Triton X-100 (Sigma-Aldrich) and stained using a PE-conjugated, IgG1 mouse anti-human IL-1β antibody (R&D Systems, clone #8516, Cat No. MAB201-100).

**Western blot analysis**. After stimulation platelets were immediately lysed on ice using cell lysis buffer (Cell Signaling Technology, USA). Protein concentration was determined using BCA (bicinchoninic acid) protein assay reagent kit according to the manufacturer's protocol. Equal amounts of protein were separated by gel electrophoresis (SDS-PAGE) and blotted onto a nitrocellulose membrane. Membranes were blocked by incubation 5% (w/v) BSA in Tris-buffered saline with Tween (TBST) for 30 min prior to incubation at 4 °C overnight with indicated antibodies. Rabbit anti-human phospho-Syk Tyr525/526 (clone C87C1, Cat No. 2710), anti-human Syk (polyclonal, Cat No. 2712), anti-human p-Src Tyr416 (clone D49G4, Cat No. 6943), anti-human Src (clone 36D10, Cat No. 2109), and anti-beta-actin (clone D6A8, Cat No. 8457) were from Cell Signaling Technology. Anti-phospho-PLCγ2 (clone #1016D, Cat No. MAB74542) was from R&D Systems. After washing with TBST the membrane was incubated with a horseradish peroxidase conjugated secondary antibody for 1 h at room temperature. Enzymatic activity was detected with a chemiluminescence detection kit according to the supplier's protocol and recorded with a digital camera (Hamamatsu). Densitometric analysis of the blots was carried out digitally using HOKAWO Software.

**Platelet aggregation**. Platelet aggregation in PRP was carried out by optical aggregometry. PRP was incubated with LL-37 or vehicle and aggregation was started by adding ADP, collagen, or TRAP under continuous stirring at 1000 r.p.m. at 37 °C and measured in a two-channel-aggregometer (ChronoLog 490-2D, Havertown, USA). Percentage of maximal platelet aggregation was analyzed 6 min after addition of the agonist using Aggrolink software (ChronoLog, USA).

**Platelet spreading**. Isolated human platelets (2 × 10$^7$/mL) were allowed to spread for 1 h on fibrinogen-coated µ-slides (Ibidi, Martinsried, Germany), following stimulation with LL-37 (5 µmol/L) or thrombin (0.5 U/mL). Non-adherent platelets were removed after washing gently and spreading platelets fixed with 1% formaldehyde, permeabilized with 0.1% triton and stained using alexa546-labeled phalloidin (Invitrogen). Spreading was analyzed by fluorescence microscopy (Axiovert 200M microscope Zeiss, Jena, Germany).

**Inhibitors of platelet signaling pathways**. Each inhibitor was added 20 min before stimulation with LL-37 or CRAMP. BAPTA (inhibitor of intracellular calcium release); U-73122 (phospholipase C inhibitor); GF109203X (Protein kinase C inhibitor); SB203580 (p38-MAPK inhibitor), MG101 (calpain inhibitor); wortmannin (phosphoinositide 3-kinase inhibitor); Boc-MLF (FPR1-receptor inhibitor); WRW4 (FPR2-receptor inhibitor); and A438079 (P2X7 receptor inhibitor) were all obtained from Tocris (Bristol, UK). Pertussis toxin and cholera toxin (inhibitors of G-protein signaling) and Tirofiban (GPIIb/IIIa inhibitor) were from Sigma. Abciximab was from Elly Lilly (USA). Dasatinib (inhibitor of Src-family kinases), R406 (Syk Inhibitor), and Stattic (STAT3-Inhibitor) were from Selleckchem (Houston, USA). Anti-mouse GPVI antibody was from Emfret (clone JAQ1, Cat No. M011-0). The inhibitory monoclonal antibody HGP5C4 directed against human GPVI and the respective isotype control antibody RmC7H8 were generated by immunization of Lou/C rats with an adenovirally expressed human GPVI-Fc fusion protein. The latter represents a soluble form of GPVI with the extracellular domain of human GPVI fused to the human Fc domain. 4C9 and 5C4 monoclonal antibodies (immunoglobulin G1 subtype) specifically bound to GPVI-Fc but not control Fc[65]. RmC7H8, raised in rats against an irrelevant human antigen, served as control monoclonal antibody (mAb).

**Isolation of neutrophils**. Human and murine neutrophils were isolated from anticoagulated whole blood of healthy adult volunteers or mouse BM. Isolation of the neutrophils was carried out using a discontinuous isotonic PercollTM gradient (52/64/72%) and centrifuged at $1000 \times g$ for 30 min. PMNs were collected from the 64/72% interface, washed in PBS. After isolation, human neutrophils ($2 \times 10^6$/mL) were directly suspended in adhesion medium (HBSS supplemented with 20 mM HEPES, 0.25% BSA, 0.1% glucose, 1.2 mM $Ca^{2+}$, and 1.0 mM $Mg^{2+}$). Murine neutrophils were cultivated for 24 h in RPMI1640 medium supplemented with 20% WEHI-3B-conditioned medium at 37 °C and 5% $CO_2$. Neutrophil viability evaluated by trypan blue exclusion test was >99% for human neutrophils and >95% for mouse neutrophils, respectively.

**Platelet–neutrophil aggregates formation and neutrophil activation in vitro**. The effects of cathelicidin-induced platelet activation on platelet–neutrophil interactions in vitro was examined using platelets and neutrophils isolated from human volunteers or mice as described above. Platelets were treated with 5 μmol/L LL-37 for 20 min and labeled with FITC-conjugated anti-human or anti-mouse CD41 antibody (Abd-Serotec or eBiosience, respectively). A total of $10^7$ platelets were co-incubated with $10^6$ neutrophils in a volume of 400 μL neutrophil adhesion medium (see above). In some experiments a blocking mouse anti-human CD62P antibody (BioLegend, clone AK4, Cat No. 304902) was added. Analysis of platelet–neutrophil aggregates was carried out by flow cytometry. Neutrophils were identified by forward and sideward scatter characteristics, and aggregates expressed as the percentage of platelet (CD41-FITC)-positive neutrophils. In a separate set of experiments neutrophil activation upon platelet co-incubation was assessed with a PE-conjugated anti-human CD11b antibody (BD Biosciences, clone ICRF44, Cat No. 557321) and a PE-conjugated anti-human CD62L antibody (BD Biosciences, clone DREG56, Cat No. 555544). In a separate set of experiments neutrophil intracellular ROS generation was determined using 2′-7′-dichlorodihydrofluorescein diacetate (DCFH-DA, Invitrogen, USA). Cells were incubated with 5 μg/mL (final concentration) DCFH for 20 min at 37°C and washed afterwards. Fluorescence was measured by flow cytometry (BD LSR Fortessa, BD Biosciences, USA).

**NET formation in vitro**. NET formation of freshly isolated human neutrophils was analyzed in a model of static adhesion on fibrinogen-coated μ-slide eight-well chambers (Ibidi). Platelets were isolated as described above and stimulated wit LL-37 (10 μmol/L), scrambled control peptide (10 μmol/L), or GPVI receptor activating antibody HGP4C9 (1 μg/mL) for 20 min. Afterwards they were washed in buffer, added to neutrophils and co-incubated for 1 h at 37 °C. Stimulation of neutrophils with TNFα (Peprotech) and PMA (Sigma) served as positive controls to induce NET formation. After stimulation the non-adherent cells were gently removed by washing. The adherent cells were then fixed using 4% PFA and stained with anti-human MPO antibody (Abcam), anti-human histone H3 citrulline antibody (7C10) (Covalab, France) and respective secondary antibody and DAPI to visualize NET formation. Imaging was conducted by fluorescence microscopy (Axiovert 200M microscope Zeiss, Jena, Germany) and quantitative analysis was carried out by manual counting of NET-structures per visual field of 3-5 randomly defined ROIs per sample.

**Platelet preparation for cremaster experiments**. Platelets were isolated from WT C57Bl/6 mice. Platelets were labeled with Rhodamine (0.05%; 50 μL/mL, Sigma) for intravital studies and with anti-GPIb-X649 (1:200; non-blocking antibody, Emfret, Germany) for post-imaging FACS-analysis. Labeling was carried out in PRP for 30 min in the presence of prostaglandin (0.5 μg/mL) to avoid activation, and platelets were then centrifuged at $400 \times g$ for 5 min. Isolated platelets were treated with 20 μmol/L CRAMP for 20 min. CRAMP was washed out by another centrifugation step at $400 \times g$ for 5 min and platelets were resuspended in buffer.

Platelets ($10^7$ per mouse) were injected via a tail vein catheter to a recipient mouse with exposed cremaster muscle for intravital imaging.

**Intravital microscopy in the cremaster muscle**. Trauma-induced inflammation of the cremaster muscle was analyzed in anesthetized lysozyme 2 (LysM)-eGFP mice, in which mostly neutrophils express the GFP, allowing imaging of neutrophil behavior and interaction with platelets. The cremaster muscle was exteriorized following incision of the scrotum. The muscle was opened through a longitudinal incision, immobilized on a customized intravital microscopy stage and constantly superfused with warmed bicarbonate-buffered saline, equilibrated with 5% $CO_2$ in $N_2$[66]. Analysis of platelets and leukocytes was carried out by analyzing mean rolling velocities, number of adherent cells per mm², extravasated neutrophils and tethering platelets were determined at different time points after injection of platelets (0, 10, 30 min) using intravital epifluorescence microscopy (Olympus BX51WI microscope, water immersion objective ×20, 0.95 numerical aperture, Olympus). All scenes were recorded using a CCD camera (model CF8/1 HS, Kappa) and virtual dub software for later off-line analysis. Blood flow velocity was measured using a dual-slit photodiode device (Circusoft Instrumentation, Hockessin, Germany). During the entire observation, the cremaster muscle was superfused with thermo-controlled (35 °C) bicarbonate-buffered saline. Postcapillary venules under observation ranged from 20–40 μm in diameter. Microvascular parameters (venular diameter, venular vessel segment length) were determined using Fiji software. Neutrophil rolling, adherence, and extravasation were analyzed. Those platelets that were not moving during at least one single image of the movie (exposure time 40 ms) were considered as tethering platelets. Analysis of mouse blood cell counts was carried out using an Idexx Procyte Dx hematology analyzer (Idexx Europe, Hoofddorp, the Netherlands).

**Platelet–neutrophil aggregates ex vivo**. Platelet–neutrophil aggregates formation was analyzed in whole blood drawn from the inferior vena cava after intravital imaging of the mouse cremaster. Erythrocytes were lysed using FACS lysing solution according to manufacturer guidelines (BD Biosciences, USA). Platelet–neutrophil aggregates were analyzed by flow cytometry. Neutrophils could be identified by their green fluorescence in LysM-eGFP mice, platelets were identified by labeling with the non-blocking GPIb-antibody X649 (Emfret, Germany).

**Mouse model of ALI**. Anaesthetized mice received an intratracheal injection of 2 μL/g body weight of 0.1 mol/L HCl (pH 1.5) and mice were killed 4 h later. BAL was performed immediately before euthanasia. Therefore, the trachea was dissected and cannulated (PortexFineBore Polythene Tubing, 0.28 mm inner diameter/0.61 mm outer diameter, Smiths Medical International, Keene, NH). $5 \times 0.5$ mL PBS was injected and withdrawn. Thereafter, the ribcage was opened by a midline incision and the pulmonary vasculature was rinsed with 15 mL ice-cold PBS with 0.5 mmol/L EDTA after cutting the inferior cava vein to facilitate exsanguination. The lungs were removed, minced and digested with Liberase (1:20; 25 mg Liberase RI/mL aqua, Roche, Mannheim, Germany). Digested lungs were passed through a cell strainer (70 μm; MiltenyiBiotec GmbH, Bergisch Gladbach, Germany) and the resulting single cell suspension was centrifuged for 5 min at $300 \times g$. The pellets were resuspended in 1 mL Hank's balanced salt solution with 0.3 mmol/L EDTA and 0.1% BSA. Similarly, the BAL fluid was centrifuged for 5 min at $300 \times g$ and cell pellets were resuspended.

**Flow cytometry analysis of neutrophils in ALI**. Resuspended cells of digested lungs and BAL fluid were labeled with PerCP-Cy5.5 anti-mouse Ly6G (eBioscience, Cat No. 127615), APC-Cy7 anti-mouse CD45 (eBioscience, Cat No. 103115), and PE anti-Mouse CD41 (BD, Cat No. 561850). Neutrophils were identified as CD45 and Ly6G positive cells. All studies were carried out on a FACS Canto II (BD Biosciences) and data were analyzed using FlowJo software.

**Lung permeability in ALI**. FITC-Dextran (70 kDa, Sigma-Aldrich) was used to assess vascular leakage. A volume of 100 μL FITC-Dextran (30 mg/mL) were administered by tail vein injection 30 min prior to euthanasia and dye extravasation was used to assess change in vascular permeability. The fluorescence of 100 μL BAL supernatant ($Fluo_{BAL}$) and of 50 μL serum ($Fluo_{Serum}$) was measured and the volume of passaged fluid was expressed in microlitres ($V_{Perm} = (Fluo_{BAL}/100$ μL)/ $(Fluo_{Serum}/50$ μL) x BAL volume)[67].

**Histological and immunofluorescence analysis in ALI**. In some animals, one part of the right lung was fixed in formalin, embedded in paraffin wax and stained with Mayer's hematoxylin and eosin for histological examination, or placed in sucrose 30% for 12 h and then embedded in OCT for immunofluorescence staining.

**Blood oxygen saturation in ALI**. In a separate set of ALI experiments without external oxygen supplementation, blood oxygen saturation was measured by noninvasive pulse oximetry using MouseOx Plus (Starr Life Sciences, USA). The experiments were carried out in deep anesthesia as described above and were

terminated (and counted as deceased), when mice showed clinical signs of respiratory failure.

**Statistical analysis**. Data were analyzed using paired or unpaired Student's $t$-test or one-way repeated measures ANOVA (Bonferroni Correction) as appropriate to compare normally distributed variables and Mann–Whitney $U$-test or ANOVA on ranks (Dunn's method) when normal distribution was not given. All data are expressed as mean ± SEM. To compare survival distributions log-rank test (Mantel-Cox test) was used. Differences were considered significant when the error probability was $P < 0.05$.

**Data availability**. The data that support the findings of this study are available from the corresponding author upon reasonable request.

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

## Acknowledgements

The authors thank Mohammed Ikram, Irene Schubert, Ronja Schuchert, Michael Lorenz and Sebastian Helmer for excellent technical assistance. We also thank Susanne Sauer, Nicole Blount, and Beat Jantz for animal care. We thank Elisabeth Kremmer for providing anti-human GPVI antibodies. We thank David Stegner for assistance with providing $Syk^{fl/fl}$ PF4-Cre mice. We thank Hella Thun for graphical assistance (Supplementary Fig. 9). This work was supported by the Deutsche Forschungsgemeinschaft (SFB 914 project A2 to B.W., project A10 to C.S., project B1 to M. S., projects B2 and Z01 to S.M., project B8 to O.S., as well as the SFB 1123 projects A7 and B6 to S.M. and K.S., projects A5 and B6 to O.S.), the DZHK (German Centre for Cardiovascular Research) and BMBF (German Ministry of Education and Research) to S. M., the Friedrich-Baur-Foundation to J.P. and the FöFoLe program (Förderprogramm für Forschung und Lehre) of the Ludwig-Maximilians-University Munich (J.P. and A. M.). J.P. is supported by a Gerok position of the SFB 914.

## Author contributions

J.P., T.C., and C.S. designed the study, carried out experiments and wrote the paper; A.E., C.E., E.G., A.M., J.G., P.S., A.T., H.I-A., K.S., T.P., T.S., L.T.W, and D.H. carried out experiments; D.H. provided human lung tissue specimen; P.S., J.N., A.S., and S.M. provided human thrombus material; M.S., B.N., B.W., and O.S. provided critical mouse models; H.M. and D.H. provided critical in vitro tools; P.S., M.S., A.S., H.M., B.W., D.H., O.S., and S.M. discussed data and revised the manuscript.

## Additional information

**Competing interests:** The authors declare no competing interests.

