## [Peer Review File · Nature Communications]

Reviewers' comments:

Reviewer #1 (Remarks to the Author):

This is an interesting, well written report showing a novel role of neutrophil derived cathelicidin in arterial thrombosis and acid induced lung injury. On a mechanistic level the authors show that cathelicidin induces selective platelet activation via GPVI and Syk signaling, leading to P-selectin exposure. In the cremaster muscle model, this promotes formation of platelet-neutrophil aggregates, increased neutrophil rolling, adhesion and tissue infiltration. For the most part the studies are well done and clearly described. There are a few comments:

- 1) The documentation of NET formation could be more solid e.g. by measurement of additional markers such as citrullinated histones. Fig 1a shows a single image suggesting a net in a human arterial thrombus. How common was this observation? Were nets identified in the mouse arterial thrombus model? How about in the ALI model?
- 2) The dose-responses for cathelicidin on platelet p-selectin, CD40L etc seem peculiar with a response only seen at 5uM. Do the authors have any explanation for this apparent threshold effect? Is there any idea of the relevant in vivo concentration? Please discuss. In the experiments where a single dose of cathelicidin was used the concentration is not indicated in the Fig legend.
- 3) The evidence for neutrophil activation (Fig 4C) is modest. Can the authors provide further support for the concept of a positive feedback from platelets to neutrophils and any evidence of the relevant mechanisms on the neutrophil side?
- 4) In the ALI model, is there a mortality benefit of cathelicidin deficiency?
- 5) Please provide information to support the specificity of the cathelicidin antibodies used in human and mouse tissues.

Reviewer #2 (Remarks to the Author):

The manuscript by Pirchner and Czermak et al describes a novel mechanism in the cross talk of neutrophils and platelets. Neutrophils release LL-37, an antimicrobial peptide stored within cytoplasmic granules, to activate platelets, which in turn stimulate neutrophils to release pro-thrombotic extracellular traps (NETs). Consequently, the proposed mechanism forms a vicious cycle that propels immunothrombosis. The manuscript is well written, the experiments are well performed, and the data are presented in a conclusive manner. However, some concerns require attention justify the proposal of the novel mechanism:

Major concerns:

- The binding of LL-37 from neutrophils to platelets is a key element of the proposed mechanism. However, the manuscript lacks a direct evidence for such scenario. The authors use mostly purified LL-37 to study the effects on platelets. The study would profit from data that illustrates LL-37 binding to platelets in vivo. Such data should be obtained from an experiment that omits the use of purified LL-37.
- The novelty or conclusions of the histological data is not clear to this Reviewer: Figs. 1a/b show stainings from human and murine thrombi. The LL-37 staining shows a granular pattern, which would indicate that LL-37 is localized within the neutrophils. Thus staining for LL-37 and MPO detects neutrophils in thrombi. For a conclusive study, the authors may also explain why no

CRAMP-positive platelets (CD41) are detected. In addition, the "NET"-structure lacks largely and LL-37 or MPO. This suggests that the shown picture may represent a crawling neutrophil with an elongated nucleus. Furthermore, statements such as "cathelicidins are highly abundant in arterial thrombi" would be strengthened by a quantification of cathelicidins along with a reference marker (e.g. platelets, fibrin).

- The data on LL-37 induced NET-formation is not convincing (Fig. 4d). In fact, the condition LL-37+IgG shows an MPO-positive string, which lacks DNA (DAPI). This suggests that the authors may not detect NETs, but "neutrophil cytonemes" (see Fig. 4, EMBO Rep. 2013 Aug; 14(8): 726–732.)

Minor concerns:

- A comprehensive base line characterization hemostatic parameters (platelet function, clotting assays) of CRAMP-/- mice as well as mice from BM-transplants should be performed to exclude the possibility that the observed in vivo effects are due to e.g. a reduced platelet count or prolonged plasma clotting time.

- Fig. 4f: Samples treated with vehicle lack error bars. For statistical analysis, the variation among these samples needs to be taken into account.

Reviewer #1:

This is an interesting, well written report showing a novel role of neutrophil derived cathelicidin in arterial thrombosis and acid induced lung injury. On a mechanistic level the authors show that cathelicidin induces selective platelet activation via GPVI and Syk signaling, leading to P-selectin exposure. In the cremaster muscle model, this promotes formation of platelet-neutrophil aggregates, increased neutrophil rolling, adhesion and tissue infiltration. For the most part the studies are well done and clearly described. There are a few comments.

We thank the reviewer for these positive remarks. We performed several additional experiments to address the comments raised.

1) The documentation of NET formation could be more solid e.g. by measurement of additional markers such as citrullinated histones. Fig 1a shows a single image suggesting a net in a human arterial thrombus. How common was this observation? Were nets identified in the mouse arterial thrombus model? How about in the ALI model?

We thank the reviewer for this comment. Compared to the stimulation experiments performed in 2D *in vitro* (Fig. 5f, g), NETs are very difficult to assess in a 3D thrombus structure, because due to tissue sectioning they are often missed out on. As suggested, we carried out citrullinated histone H3 (citH3) immunofluorescence staining of thrombus specimen and of lung tissue in the ALI model. citH3⁺ neutrophils were identified in human and mouse arterial thrombi (new Suppl. Fig. 4). In thrombi of *Cramp*^{+/+} BM chimeric mice, approximately 60% of Ly6G⁺ cells were positive for citH3⁺. The results were not significantly different to thrombi of *Cramp*^{-/-} BM chimeric mice (new Suppl. Fig. 4).

In lung tissue of *Cramp*^{+/+} BM chimeric mice with ALI, citH3⁺ neutrophils were also present albeit less abundant (5-6% of Ly6G⁺ cells) than in arterial thrombi. Absence of CRAMP reduced the overall number of neutrophils accumulating in ALI lungs and also the number of citH3⁺ neutrophil (new Fig. 7j). However, the relative amount of neutrophils with citH3 staining was not altered (new Fig. 7i-k). The findings suggest that cathelicidins are not required for the priming of neutrophils towards NETosis *in vivo* in the conditions analyzed.

Notwithstanding the above, cathelicidins can associate with extracellular DNA released by neutrophils¹, and could potentially be presented to nearby platelets to foster thrombo-inflammatory conditions. Further, cathelicidin-activated platelets elicit neutrophil activation in various ways, in addition to NET formation observed *in vitro*. LL-37 pre-stimulated platelets increased neutrophil CD11b expression, L-Selectin shedding and promoted neutrophil production of reactive oxygen species (Fig. 5c and new Fig. 5d,e), which could potentially mediate the effects of LL-37/CRAMP on arterial thrombus formation and lung injury.

2) The dose-responses for cathelicidin on platelet p-selectin, CD40L etc seem peculiar with a response only seen at 5uM. Do the authors have any explanation for this apparent threshold effect? Is there any idea of the relevant in vivo concentration? Please discuss. In the experiments where a single dose of cathelicidin was used the concentration is not indicated in the Fig legend.

We thank the reviewer for this comment. We carried out an extended dose-response analysis, which is now shown in the revised figures 2a. 2.5 μM LL-37 leads to modest albeit significant effects on platelet P-Selectin expression. However, 5 μM LL-37 was the lowest concentration that elicited a robust platelet activation response *in vitro*. Higher concentrations did not further enhance platelet activation.

The *in vivo* concentration of cathelicidins has been measured in some body fluids²⁻⁵. The estimated levels of 1 $\mu\text{mol/L}$ and 10 $\mu\text{mol/L}$ for LL-37 and CRAMP, respectively, are in the range of the concentrations used in the present study. In the discussion section of the revised manuscript, we now discuss this issue as follows: “*The physiological concentrations of cathelicidins that prevail in vivo are incompletely understood. Analysis of bronchoalveolar lavage fluid and samples taken from different mucosal sites estimates concentrations to be around 1 $\mu\text{mol/L}$ and 10 $\mu\text{mol/L}$ for LL-37 and CRAMP, respectively²⁻⁴, which are much higher than the concentrations reached in plasma⁵. The in vitro concentrations applied in this study (5 $\mu\text{mol/L}$ LL-37, and 20 $\mu\text{mol/L}$ CRAMP) are in this order of magnitude. Further, in our histological analysis we found that cathelicidins were abundant in the inflamed lung and in arterial thrombi, which is in line with the local enrichment of LL37/CRAMP within other tissues²⁻⁵. However, measuring in vivo levels of cationic peptides has been challenging⁶, and the precise concentrations reached locally at the site of tissue injury remain to be determined.*”

We added the concentration of cathelicidin used in the single-dose experiments.

3) The evidence for neutrophil activation (Fig 4C) is modest. Can the authors provide further support for the concept of a positive feedback from platelets to neutrophils and any evidence of the relevant mechanisms on the neutrophil side?

In Fig. 5c (previous Fig. 4c), we showed that neutrophil CD11b surface expression was increased after incubation with LL-37 activated platelets. In the revised manuscript, we carried out additional experiments to address neutrophil production of reactive oxygen species and L-Selectin shedding. We found that LL-37-activated platelets induced significant ROS production (new Fig. 5d) as well as shedding of CD62L from the surface of neutrophils (new Fig. 5e). Thus, we provide further evidence that cathelicidin-activated platelets induce neutrophil activation.

4) In the ALI model, is there a mortality benefit of cathelicidin deficiency?

We thank the reviewer for this very interesting question. We carried out additional ALI experiments in mice to determine the impact of cathelicidin on mortality. Indeed, absence of CRAMP in bone marrow chimeric mice was associated with a survival benefit in the lung inflammation model. The mortality data are supported by non-invasive measurements of blood oxygen saturation. The new data is included in figures 7s and 7t.

5) Please provide information to support the specificity of the cathelicidin antibodies used in human and mouse tissues.

We routinely used isotype-matched control antibodies to determine the specificity of the cathelicidin antibodies. Controls for immunohistochemistry are shown in figures 1a and 1c. Immunofluorescence control stainings are shown in supplementary figures 1 and 2. Further, we carried out additional stainings in the spleen of CRAMP-deficient mice to proof the absence of unspecific antibody binding (new supplementary figure 2b).

Reviewer #2:

The manuscript by Pirchner and Czermak et al describes a novel mechanism in the cross talk of neutrophils and platelets. Neutrophils release LL-37, an antimicrobial peptide stored within cytoplasmic granules, to activate platelets, which in turn stimulate neutrophils to release pro-thrombotic extracellular traps (NETs). Consequently, the proposed mechanism forms a vicious cycle that propels immunothrombosis. The manuscript is well written, the experiments are well performed, and the data are presented in a conclusive manner. However, some concerns require attention justify the proposal of the novel mechanism.

We thank the reviewer for these positive comments.

The binding of LL-37 from neutrophils to platelets is a key element of the proposed mechanism. However, the manuscript lacks a direct evidence for such scenario. The authors use mostly purified LL-37 to study the effects on platelets. The study would profit from data that illustrates LL-37 binding to platelets in vivo. Such data should be obtained from an experiment that omits the use of purified LL-37.

We agree with the reviewer on this challenging question. We addressed the comment by carrying out additional experiments both in vitro and in vivo.

1) To determine the binding of cathelicidins to the platelet-rich thrombus, we collected human and mouse arterial thrombi after thrombus formation in vivo, i.e. in patients with myocardial infarction as well as in a mice after carotid artery injury. Immunohistochemical analysis demonstrated that staining for both LL-37 and CRAMP is positive in areas of the thrombus, in which only few leukocytes are present (Fig. 1a,c). These findings indicate, without use of purified peptides, that cathelicidins associate with platelets in arterial thrombi in both humans and mice.

2) To support the notion that cathelicidins bind to platelets in vivo, we harnessed a custom made version of the CRAMP peptide that was tagged with 5-Carboxyfluorescein (5-FAM). In separate experiments, we injected either 5-FAM-CRAMP or a 5-FAM-labeled scrambled control version of the peptide into mice and induced arterial thrombosis. Association of the peptide with platelets at the site of injury was determined by intravital microscopy. 5-FAM-CRAMP but not the control peptide readily accumulated at the site of platelet-thrombus formation. The data is included in figure 1e and in video files as supplementary movies 1 and 2.

3) Finally, we carried out flow cytometry analysis of cathelicidin binding to platelets. We incubated the 5-FAM-tagged LL-37 peptide with isolated human platelets, and analyzed binding of the peptide after washing. Comparable to the results observed in vivo, LL-37, but not the scrambled 5-FAM-tagged peptide, showed significant binding to platelets (new Fig. 1f).

Taking the results of these three separate experiments together, we provided evidence that LL-37/CRAMP binds to platelets. Although we made use of purified CRAMP in two of the experiments performed, we believe that the experiments support these conclusions.

The novelty or conclusions of the histological data is not clear to this Reviewer: Figs. 1a/b show stainings from human and murine thrombi. The LL-37 staining shows a granular pattern, which would indicate that LL-37 is localized within the neutrophils. Thus staining for LL-37 and MPO detects neutrophils in thrombi. For a conclusive study, the authors may also explain why no CRAMP-positive platelets (CD41) are detected. In addition, the “NET”-structure lacks largely LL-37 or MPO. This suggests that the shown picture may represent a crawling neutrophil with an elongated nucleus. Furthermore, statements such as “cathelicidins are highly abundant in arterial thrombi” would be strengthened by a quantification of cathelicidins along with a reference marker (e.g. platelets, fibrin).

We agree with the reviewer and apologize for the confusing histological data. We carried out additional stainings of mouse and human arterial thrombi.

Neutrophils are the major source of cathelicidins, and LL-37 is highly enriched within these cells as shown by immunofluorescence staining and immunohistochemistry (new Fig. 1a,b and Suppl. Fig. 1a,b). Due to the strong fluorescent signal of neutrophils and due to thrombus autofluorescence, it is difficult to determine the association of secreted cathelicidins with platelets (CD41) across the thrombus using immunofluorescence analysis. We therefore carried out immunohistochemistry in both human and mouse arterial thrombi. We found that, in addition to the strong dark-brown signal provided by neutrophils, LL-37 also associated with areas of arterial thrombi in which leukocytes were mostly absent (new fig. 1a). Similar findings were made for immunohistochemistry of CRAMP in mouse arterial thrombi (new fig. 1c).

The immunohistochemistry of human and mouse arterial thrombi (new Fig. 1a,c) indicates that cathelicidins are abundant in arterial thrombi. The term “highly” was omitted. Further, we performed concomitant immunofluorescence staining for LL-37 and CD41 (as reference) supporting the histological findings outlined above (new Suppl. Fig. 2b).

It is very difficult to assess NET structures in a 3D thrombus, because they are often missed out on due to tissue sectioning. This is in contrast to the neutrophil stimulation experiments performed in 2D *in vitro* (Fig. 5f, g). To determine neutrophil activation and priming towards NETosis, we carried out immunofluorescence staining of citrullinated histone H3 (citH3) in thrombus specimen. citH3⁺ neutrophils were identified in human and mouse arterial thrombi (suppl. fig. 4). Further, both MPO and LL-37 are associated with citH3⁺ neutrophils (suppl. fig. 4a).

The data on LL-37 induced NET-formation is not convincing (Fig. 4d). In fact, the condition LL-37+IgG shows an MPO-positive string, which lacks DNA (DAPI). This suggests that the authors may not detect NETs, but “neutrophil cytonemes” (see Fig. 4, EMBO Rep. 2013 Aug; 14(8): 726–732.)

We thank the reviewer for this comment and apologize for the confusing image presentation. In the respective image of figure 5g (previous fig. 4d), NET formation is indicated by the presence of a string that stained positive for both DNA (DAPI) and MPO. The single images of the same picture indicate the staining with DAPI (white, upper row) and MPO (red, middle row). In the bottom row, the merged image of DAPI (blue) and MPO (red) is displayed, however, the red is rather dominant. Please note that similar strings appeared when neutrophils were incubated with platelets that were pretreated with LL-37 (2nd column) or GPVI (5th column). We revised this figure to more clearly display the presence of NETs. We marked the DAPI-positive structure with an arrowhead throughout all rows. Further, on the left side of the image we now indicate the staining performed in respective colors.

Further, we carried out co-staining of MPO, DAPI and citH3 in these conditions to provide additional evidence for NET formation. The new data was included in figure 5f.

Minor concerns:

A comprehensive base line characterization hemostatic parameters (platelet function, clotting assays) of CRAMP^{-/-} mice as well as mice from BM-transplants should be performed to exclude the possibility that the observed *in vivo* effects are due to e.g. a reduced platelet count or prolonged plasma clotting time.

We thank the reviewer for this comment. We carried out a baseline characterization of hemostatic parameters in wildtype and *Cramp^{-/-}* BM chimeric mice. We tested blood cell counts, platelet activation markers and bleeding time. We also carried out ROTEM analysis in order to provide a more detailed characterization of clot formation. No differences were found between groups, indicating that the observed *in vivo* effects were not due to alterations in hemostatic parameters.

Fig. 4f: Samples treated with vehicle lack error bars. For statistical analysis, the variation among these samples needs to be taken into account.

We apologize for this mistake. We added error bars to take their variation into account.

References

1. Lande R, Gregorio J, Facchinetti V, Chatterjee B, Wang YH, Homey B, Cao W, Wang YH, Su B, Nestle FO, Zal T, Mellman I, Schroder JM, Liu YJ and Gilliet M. Plasmacytoid dendritic cells sense self-DNA coupled with antimicrobial peptide. *Nature*. 2007;449:564-9.
2. Frigimelica E, Bartolini E, Galli G, Grandi G and Grifantini R. Identification of 2 hypothetical genes involved in *Neisseria meningitidis* cathelicidin resistance. *The Journal of infectious diseases*. 2008;197:1124-32.
3. Bowdish DM, Davidson DJ, Lau YE, Lee K, Scott MG and Hancock RE. Impact of LL-37 on anti-infective immunity. *Journal of leukocyte biology*. 2005;77:451-9.
4. Schaller-Bals S, Schulze A and Bals R. Increased levels of antimicrobial peptides in tracheal aspirates of newborn infants during infection. *American journal of respiratory and critical care medicine*. 2002;165:992-5.
5. Malm J, Sorensen O, Persson T, Frohm-Nilsson M, Johansson B, Bjartell A, Lilja H, Stahle-Backdahl M, Borregaard N and Egesten A. The human cationic antimicrobial protein (hCAP-18) is expressed in the epithelium of human epididymis, is present in seminal plasma at high concentrations, and is attached to spermatozoa. *Infection and immunity*. 2000;68:4297-302.
6. Bowdish DM, Davidson DJ and Hancock RE. A re-evaluation of the role of host defence peptides in mammalian immunity. *Current protein & peptide science*. 2005;6:35-51.

REVIEWERS' COMMENTS:

Reviewer #1 (Remarks to the Author):

The revised manuscript is substantially improved and responsive to the previous criticisms.

Reviewer #2 (Remarks to the Author):

No further comments